# EFFICIENT OPEN-SET TEST TIME ADAPTATION OF VISION LANGUAGE MODELS

## ABSTRACT

In dynamic real-world settings, models must adapt to changing data distributions, a challenge known as Test Time Adaptation (TTA). This becomes even more challenging in scenarios where test samples arrive sequentially, and the model must handle open-set conditions by distinguishing between known and unknown classes. Towards this goal, we propose ROSITA, a novel framework for Open set Single Image Test Time Adaptation using Vision-Language Models (VLMs). To enable the separation of known and unknown classes, ROSITA employs a specific contrastive loss, termed ReDUCe loss, which leverages feature banks storing reliable test samples. This approach facilitates efficient adaptation of known class samples to domain shifts while equipping the model to accurately reject unfamiliar samples.Our method sets a new benchmark for this problem, validated through extensive experiments across diverse real-world test environments.. Our code is anonymously released at `https://github.com/anon-tta/ROSITA.git`

## 1 INTRODUCTION

Over the past decade, substantial advancements have been achieved in various computer vision tasks Deng et al. (2009); Ren et al. (2015); He et al. (2017); Everingham et al. (2010). However, these achievements are predominantly realized under the assumption that both training and test data originate from the same distribution. In contrast, the real world is dynamic and ever-changing, making such assumptions often untenable. Distribution gaps between training and test data manifest in diverse forms Hendrycks & Dietterich (2019); Peng et al. (2019b), including domain shifts and semantic shifts. Domain shifts emerge from variations in lighting, weather, camera specifications, or geographical locations between the train and test datasets. Semantic shifts occur when a model, initially trained on a specific set of classes, encounters previously unseen classes during testing. Hence, navigating deep learning models through these dynamic test environments is imperative.

Researchers have been tackling the robustness of models to domain shifts, by diving into paradigms like Unsupervised Domain Adaptation Ganin et al. (2016), Source-Free Domain Adaptation Liang et al. (2020); Yang et al. (2022). More recently, the problem of Test Time Adaptation (TTA) Wang et al. (2021); Schneider et al. (2020); Niu et al. (2022) and Continuous Test Time Adaptation (CTTA) Döbler et al. (2023) has come to the forefront. TTA is characterized by three key factors: *(1) No access to source data; (2) No ground truth labels for test data; (3) An online adaptation scenario where the model encounters test samples only once*, reflecting the online nature of real-world.

Another facet of distribution gaps lies in semantic shifts Li et al. (2023); Lee et al. (2023). While TTA methods have predominantly focused on closed-set scenarios, the real world seldom operates within such constraints. A classic example is that of autonomous driving Wang et al. (2022), where models trained for specific geographical locations are deployed elsewhere. For instance, a model trained to recognize only vehicles commonly seen in urban areas—such as *car, truck, motorcycle*—may incorrectly classify a *bicycle* as a *motorcycle* when deployed in rural settings. In these new environments, the model must be able to *identify elements that are not relevant to its training as unknown, rather than misclassifying them as part of the known set of categories*. This underscores the importance of *Open Set Adaptation*. Though this has only recently been explored in the context of TTA Li et al. (2023); Lee et al. (2023), current TTA and CTTA methods Wang et al. (2021); Döbler et al. (2023) generally rely on *accumulating a batch of images* to update the model, which may not be feasible in scenarios where test samples arrive individually. This highlights the growing need for efficient *Single Image Test Time Adaptation* methods.

Parallel to the recent advances in TTA, there has been tremendous progress in the development of large scale Vision Language Models (VLM) like CLIP Radford et al. (2021). Having trained on large scale web scrapped image-text pairs, these VLMs Radford et al. (2021) have demonstrated impressive zero shot generalization capabilities, making it a natural candidate for TTA. Recently, Shu et al. (2022); Samadh et al. (2023); Karmanov et al. (2024) have shown that these VLMs can be adapted on each image during inference, further improving the zero shot generalization performance.

Current VLM based works Shu et al. (2022); Karmanov et al. (2024) address Single Image TTA in closed-set setting and do not explicitly handle open set scenarios. Recent CNN based open-set TTA works Li et al. (2023); Lee et al. (2023) operate on batches of test images. In this work, we address both the challenges and establish a benchmark for *Open set Single Image Test Time Adaptation using VLMs*. We refer to the classes of interest with respect to a particular downstream classification task (say 10 classes of CIFAR-10) as *desired* classes and the rest as *undesired* classes (say 10 digits of MNIST). In such scenarios, it is necessary to filter out undesired class samples, preventing them from negatively impacting model adaptation during test-time adaptation (TTA). To achieve this, we employ a Linear Discriminant Analysis (LDA) Fisher (1936); Li et al. (2023)-based class identifier, which first determines whether a test sample belongs to a desired or undesired class. Samples identified as belonging to the desired classes are then classified accordingly into one of the desired classes. The challenges we address are twofold: (1) Enabling TTA of VLMs where samples arrive sequentially, and (2) handling open-set scenarios where test samples may belong to either Desired or Undesired classes. To tackle these challenges, we utilize **Re**liable samples to differentiate **D**esired vs **U**ndesired classes through a **C**ontrastiv**e** loss, termed ReDUCe, within the framework of **O**pen set **S**ingle **I**mage **T**est time **A**daptation (**ROSITA**). Our contributions are summarized as follows:

- To the best of our knowledge, we are the first to tackle the challenging and realistic problem of *Open set Single Image Test Time Adaptation using VLMs*, setting a new benchmark.
- We provide a comprehensive analysis on *continuous adaptation of VLMs* during test time and identify LayerNorms to be the optimal set of parameters to adapt the model.
- Our framework, **ROSITA**, adapts a VLM to recognize desired class samples with domain shifts while enabling it to effectively differentiate unfamiliar samples by saying "I don't know." This distinction between desired and undesired class samples is achieved using our ReDUCe loss, which dynamically contrasts these classes to enhance separability.
- We demonstrate the effectiveness of our method through extensive experiments across a diverse array of domain adaptation benchmarks, simulating various real-world test environments, with samples from single domain, continuous and frequently changing domains. We also experiment varying the ratio of desired and undesired class samples in the test stream.

## 2 OPEN SET SINGLE IMAGE TEST TIME ADAPTATION

### 2.1 PROBLEM SETUP

**Test stream.** The model encounters a single test sample $x_t$ at time $t$, sampled from $\mathcal{D}_t = \mathcal{D}_d \cup \mathcal{D}_u$ comprising of: (i) Desired class samples: $\mathcal{D}_d = \{x_t; y_t \in C_d\}$, with domain shift and belonging to one of the $C_d$ desired classes, for example, $C_d = \{car, bus, ..., motorcycle\}$; (ii) Undesired class samples: $\mathcal{D}_u = \{x_t; y_t \in C_u\}$, which have semantic shift (irrelevant classes) such that $C_d \cap C_u = \phi$.

**Goal.** Given a test sample $x_t$ arriving at time $t$, the goal is to be first recognize if it belongs to a desired class or not, constituting a binary classification task. If $x_t$ is identified as a desired class sample, a subsequent $|C_d|$-way classification is performed, else the prediction is "*I don't know*". In essence, the overall process can be viewed as a $|C_d| + 1$ way classification problem.

**Open set Single Image TTA scenarios.** We simulate several test scenarios inspired from the real world to evaluate the effectiveness of our method. (1) *Single domain*: We extend the standard TTA scenario where the test samples come from an unseen domain $D_d$ (say *snow* corruption of CIFAR-10C) by incorporating undesired samples $D_u$ (say MNIST). (2) *Continuously changing domains*: Here, $D_t$ changes with time as $(D_d^1 \cup D_u) \to (D_d^2 \cup D_u) \ldots \to (D_d^n \cup D_u)$, where $D_d^i$ is the $i^{th}$ domain encountered. (3) *Frequently changing domains*: Here, we significantly reduce the number of samples per domain in continuous open set TTA. Lesser the samples per domain, more frequently the domain of the test stream changes, simulating very dynamic open set test scenarios. (4) *Vary the ratio of samples from $C_d$ to $C_u$* in the test stream.

**Open set TTA in the context of pretrained VLMs.** TTA has traditionally focused on improving the performance of CNNs, where models trained on clean data struggle in unseen environments such as noisy or weather-affected conditions. However, these models are trained specifically on a dataset to recognize a desired set of classes $C_d$. Recently VLMs such as CLIP Radford et al. (2021) have demonstrated impressive zero-shot generalization performance across diverse domains without any specific retraining. Due to the contrastive pretraining of (image, text) pairs in VLM, *text-based classifiers can be obtained for free by embedding text prompts of the form "A* photo of a {class name}" through its text encoder. Image features can then be matched with text-based classifiers to perform $|C_d|$-way classification. This makes CLIP a natural candidate for TTA scenarios.

In the context of CLIP, it is non-trivial to define classes or domains as unseen, given its exposure to a vast array of visual data including variations, corruptions, and styles. In general, CLIP can be used to classify an image by making a choice from the given set of desired classes. However, it lacks the ability to explicitly say *I don't know* when presented with a sample which does not belong to the set of desired classes. Also, despite CLIP's strong zero-shot performance on clean data, its performance on corrupted/style-shifted datasets like ImageNet-C/R is still subpar (Shu et al., 2022; Karmanov et al., 2024; Zhang et al., 2024), highlighting the need for handling severe domain shifts better. This makes the problem highly relevant and worth addressing.

To address this, we establish a strong benchmark by adapting current TTA methods based on CLIP, as well as open-set TTA approaches designed for CNNs to evaluate CLIP's performance in open-set settings. Further, we introduce a novel framework called **ROSITA**, which achieves state-of-the-art results, surpassing prior methods and setting a new standard for open-set TTA.

## 2.2 BASELINES

We perform experiments using CLIP Radford et al. (2021) and MaPLe Khattak et al. (2023) backbones. CLIP consists of a Vision ($\mathcal{F}_V$) and Text ($\mathcal{F}_T$) encoder, trained using contrastive learning on image-text pairs. MaPLe backbone uses multimodal prompts to adapt CLIP for downstream tasks.

**Classification using VLMs.** Given a test image $x_t$ and a set of desired classes $C_d = \{c_1, c_2, \ldots c_N\}$, we construct the text-based classifier by first prepending each class name with a predefined text prompt $\boldsymbol{p}_T =$ "A photo of a". This forms class-specific text inputs $\{\boldsymbol{p}_T, c_i\}$, which are then passed through the text encoder to obtain text embeddings $\boldsymbol{t}_i = \mathcal{F}_T(\{\boldsymbol{p}_T; c_i\})$ for each $c_i \in C_d$. As a result, we get the text-based classifier $\{\boldsymbol{t}_1, \boldsymbol{t}_2, \ldots \boldsymbol{t}_2\}$. Finally, the class prediction is made by identifying the text embedding $\boldsymbol{t}_i$ that has the highest similarity to the image feature $f_t$.

**Desired vs Undesired Class Identifier.** In real-world, a deployed model may encounter instances from both desired and undesired classes. *We equip all methods Shu et al. (2022); Karmanov et al. (2024); Zhang et al. (2024) with an LDA based parameter-free class identifier Fisher (1936); Li et al. (2023) to reject undesired class samples.* Subsequently, the model is adapted during test time.

**Benchmark for Open-set Single Image TTA.** We adapt the single image closed-set TTA baselines ZSEval Radford et al. (2021), TPT Shu et al. (2022), PAlign Samadh et al. (2023), TDA Karmanov et al. (2024), DPE Zhang et al. (2024) for our problem setting. We also adapt TPT and PAlign for continuous model update by adapting prompts, which we refer as TPT-C and PAlign-C respectively. The test samples recognized to belong to $C_u$ are not used to update the model as they can adversely affect its performance on desired classes. We adapt two recent CNN based open-set TTA works (K+1)PC Li et al. (2023), UniEnt Gao et al. (2024) for VLMs. We refer to Li et al. (2023) as $(K+1)PC$, as they perform $(K+1)$-way Prototypical Classification. *We equip all these baselines (Appendix B) with the same LDA based desired vs undesired class identifier for fair comparison.*

We first present our preliminary analysis on continuous adaptation of VLMs. We then describe the LDA based $C_d$ vs $C_u$ class identifier Li et al. (2023) and the proposed **ROSITA** framework.

## 2.3 PRELIMINARY ANALYSIS: CONTINUOUS ADAPTATION OF VLMs

Test time adaptation methods using CNNs Wang et al. (2021); Schneider et al. (2020); Liang et al. (2020); Chen et al. (2022) successfully leverage test domain data arriving in an online manner (in batches) to continuously update the model. In this work, we study TTA of VLMs like CLIP, which has only been explored very recently Shu et al. (2022); Karmanov et al. (2024); Zhang et al. (2024) by adapting prompts independently for each image. While these methods show promise for on-the-fly

adaptation in a zero-shot framework, it is not clear whether they can leverage the online data stream to continuously update the model parameters. Based on the evidence in prior TTA works (Wang et al., 2021; Chen et al., 2022), we analyze two aspects of VLMs for the TTA task: (1) Here, we question if VLMs can be continuously adapted in a similar manner, but using only a single test image at a time; (ii) If so, are prompts (Shu et al., 2022) the best parameters to continuously update?

**Experiment.** We choose six different parameter groups: (1) Prompts, (2) LayerNorm parameters (Zhao et al., 2023), (3) Full network (4) First Attention Block of ViT (5) Last Attention Block of ViT (6) Prompts+LayerNorm(LN). We perform *single image TTA in a closed set scenario* on CIFAR-10C, by continuously adapting each of these parameter groups of CLIP, using reliable entropy loss, $L_{TTA} = \mathbf{1}(s_t > \tau)\mathcal{L}_{ent}(x_t)$, which is commonly used in several TTA methods (Wang et al., 2021; Niu et al., 2022) and VLM based prompt tuning methods TPT, PAlign. Here, $x_t$ and $s_t$ refer to the test sample and its confidence, respectively. $\tau$ is the confidence threshold used to select reliable samples Niu et al. (2022) for the model update, which we set to 0.7 in this analysis.

**Observations.** We find that continuous model adaptation can indeed improve VLMs performance based on our empirical analysis (Figure 1). (1) Using a high learning of $10^{-2}$ for any parameter group results in a severe drop in accuracy compared to the zero-shot performance of CLIP in this extreme setting of continuous single image model update. (2) The other extreme of low learning rate of $10^{-6}$ performs at par with ZSEval for all parameter groups, suggesting the model has not sufficiently changed. (3) Updating the Full Network results in an accuracy of about 10% across all learning rates, suggesting that giving the highest flexibility can cause the model to lose the inherent generalization ability of the VLM. (4) Early attention layers can potentially be updated. However, they are more sensitive to learning rate and optimizer choice (Appendix C.7). Also, Prompt updates are more expensive as the compute scales with the number of classes, making them less suitable for continuous adaptation (Appendix C.8). (5) We find that tuning the LayerNorm parameters of the Vision encoder (which account for just 0.032% of the total parameters) offers the best balance between performance and complexity.

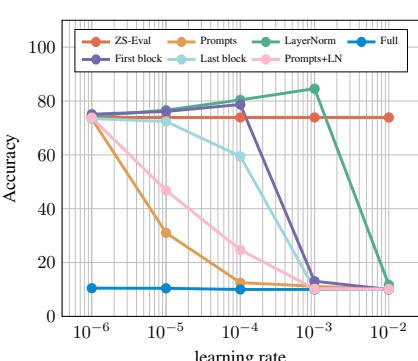

Figure 1: Accuracy on fine-tuning different parameter groups for single image TTA.

**Adapting Image encoder vs Text classifiers:** Most existing TTA approaches Schneider et al. (2020); Wang et al. (2021); Chen et al. (2022) focus on adjusting image representations for domain shifts during test time while keeping the classifiers fixed. This strategy helps retain class discriminative information. Conversely, in TPT and PAlign, the text-based classifiers which depend on learnable prompts are updated based on single images. While this does not impact zero-shot evaluation (since the model resets after each image), it can be detrimental during continuous updates.

Based on this analysis, we freeze the text-based classifiers and modify only the image representations using LayerNorm affine parameters. The rationale behind this approach is that text representations can be inherently more robust across domains. Text embeddings, often derived from a wide range of linguistic contexts, capture semantic meanings that are less susceptible to variations in visual data. Therefore, adapting the image encoder allows for more effective handling of domain shifts while retaining the class-level discriminative information from the text modality. This ensures that the model can be updated continuously without the need for resets, ultimately enhancing its performance in dynamic real open-set environments.

## 2.4 DESIRED VS UNDESIRED CLASS IDENTIFIER

Contrary to closed-set TTA setting, updating the model using all the test samples is not desirable in the open-set scenario, where test samples can come from either $C_d$ or $C_u$. It is hence imperative to equip the model with the ability to say *I don't know* by rejecting samples which do not belong to $C_d$. In the context of VLMs, we define a score ($s_t$) of a test sample to be the maximum cosine similarity with the text embeddings as given below:

$$s_t = \max_k \text{sim}(f_t, t_k); \quad k \in \{1, \dots C\} \tag{1}$$

This problem can be viewed as a binary classification problem between desired and undesired samples based on the score $s_t$. Defining a threshold to discriminate between the two can be particularly challenging in the TTA scenario as the samples are only accessible in an online manner. To circumvent this issue, following Li et al. (2023), we store the scores in a score bank $\mathcal{S}$, which is continuously updated in an online manner to store the latest $|\mathcal{S}|$ scores, approximating the latest distribution of scores of the test data. Given this, the optimal threshold can be estimated by performing 1D LDA Fisher (1936). A simple linear search over a range of thresholds is done to identify the best threshold that minimizes the variance of scores of samples from $C_d$ and $C_u$. For a threshold $\tau$, let $\mathcal{S}_d = \{s_i | s_i > \tau, s_i \in S\}$ and $\mathcal{S}_u = \{s_i | s_i < \tau, s_i \in S\}$ denote the scores of samples identified to belong to $C_d$ and $C_u$ respectively. The optimal threshold $\tau_t^*$ at time $t$ is identified as the one that minimizes the intra class variance as follows

$$\tau_t^* = \arg\min_\tau \frac{1}{|\mathcal{S}_d|} \sum_{s \in \mathcal{S}_d} (s - \mu_d)^2 + \frac{1}{|\mathcal{S}_u|} \sum_{s \in \mathcal{S}_u} (s - \mu_u)^2 \qquad (2)$$

where $\mu_d$ and $\mu_u$ are the means estimated from $\mathcal{S}_d$ and $\mathcal{S}_u$ respectively. The test sample $x_t$ is classified as desired if $s_t \geq \tau_t^*$ and undesired otherwise.s *We establish a strong benchmark for Open set Single Image TTA by equipping all the baseline methods (Section 2.2) with this simple and efficient LDA based class identifier.* In Section C.4, we demonstrate the effectiveness of this method in comparison with simple confidence thresholding. We now describe the proposed framework **ROSITA**.

## 3 PROPOSED ROSITA FRAMEWORK

Given a single test sample $x_t$ at time $t$, it is first identified as a desired or undesired class sample as described above. This is important, since, using undesired class samples can have a negative impact on model adaptation. In this work, we propose a test time objective that can leverage both desired and undesired class samples through feature banks to enhance the discriminability between them.

**Reliable samples for TTA.** We first identify a test sample $x_t$ as a *reliable desired or undesired class sample* based on its score $s_t$. As we have access to an approximate distribution of the scores as described in Section 2.4, we leverage the statistics $\mu_d$ and $\mu_u$ estimated through LDA to identify reliable samples. A test sample $x_t$ is said to be a reliable sample belonging to desired classes $C_d$ if its score $s_t > \mu_d$ and a reliable sample from any of the other classes $C_u$ if its score $s_t < \mu_u$. We leverage **Re**liable samples to differentiate **D**esired vs **U**ndesired class samples through a **C**ontrastiv**e** (**ReDUCe**) Loss for Open-set Single Image Test time Adaptation, illustrated in Figure 2.

**ReDUCe Loss.** A contrastive objective typically needs positive and negative features, the goal being to maximize the similarity between a sample and its positive (could be augmentation Chen et al. (2020) or nearest neighbours Dwibedi et al. (2021)), while minimizing its similarity with the negatives. Such objectives (Chen et al., 2020; He et al., 2020; Khosla et al., 2020; Dwibedi et al., 2021) have been extensively used to learn good image representations in a self-supervised way. While self-supervised learning assumes access to abundant data in an offline manner giving the freedom to carefully choose positives and negatives, this problem is set in an online scenario, where the test samples arrive one at a time and are accessible only at that instant. This challenging setting makes it non trivial to use objectives by Dwibedi et al. (2021). To circumvent this issue of lack of abundant test data, we propose to store two dynamically updated feature banks $\mathcal{M}_d$ and $\mathcal{M}_u$ of sizes $N_d$ and $N_u$, to store the features of reliable samples from $C_d$ and $C_u$ respectively. We propose a ReDUCe objective to contrast a reliable sample from $C_d$ by choosing its positives and negatives as the $K$ nearest neighbours from $\mathcal{M}_d$ and $\mathcal{M}_u$ respectively and vice versa for a reliable sample from $C_u$. The buffer size for $\mathcal{M}_d$ is set as $|C_d| \times K$, where $|C_d|$ is the number of desired classes and $K$ is the number of neighbours retrieved. The feature banks $\mathcal{M}_d$ or $\mathcal{M}_u$ are updated with a feature $f_t$ if it is detected as a reliable sample from $C_d$ and $C_u$ respectively.

We fetch the $K$ nearest neighbours of a reliable test sample $x_t$ from each feature bank as follows.

$$Q_d = \text{kNN}(f_t; \mathcal{M}_d); \quad Q_u = \text{kNN}(f_t; \mathcal{M}_u) \qquad (3)$$

**Case 1: Reliable sample from** $C_d$. If a test sample is identified as a reliable sample from $C_d$, we use a reliable pseudo-label loss on the sample $x_t$ and its augmentation $\tilde{x}_t$ as follows:

$$\mathcal{L}_{Re} = \mathcal{L}_{CE}(x_t, \hat{y}_t) + \mathcal{L}_{CE}(\tilde{x}_t, \hat{y}_t); \quad \hat{y}_t = \arg\max_i \text{sim}(f_t, t_i) \qquad (4)$$

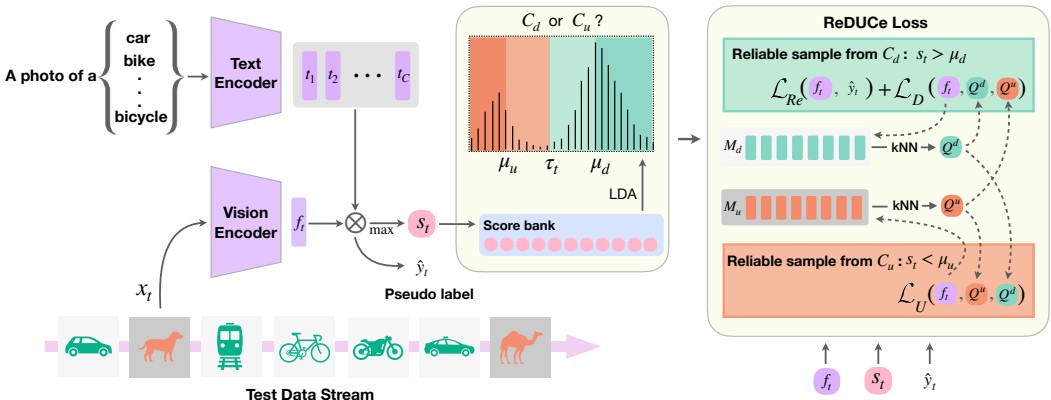

Figure 2: **ROSITA framework:** The test stream with samples from $C_d$ and $C_u$ arrive one at a time. An input image $x_t$ is recognized as a sample from $C_d$ and $C_u$ through an LDA based class identifier. Further, if a test sample is reliable, the respective feature banks are updated and the proposed ReDUCe loss is optimized to update the LayerNorm parameters of the Vision Encoder.

where sim represents cosine similarity. Further, we also propose to use a contrastive objective to enhance the clustering of desired class samples while pushing them apart from the undesired class samples.

As we aim to correctly classify the desired class samples, we select positives $z^+$ from $Q_d$ if its prediction $y^+$ matches with $\hat{y}_t$. The features $Q_u$ consisting of its kNN from $M_u$ act as its negatives. The following is the ReDUCe loss for a reliable sample from $C_d$:

$$\mathcal{L}_D = -\frac{1}{K^+} \sum_{z^+ \in Q^d} \mathbf{1}(y^+ = \hat{y}_t) \log \frac{\exp\left(\text{sim}\left(f_t, z^+\right)/\tau\right)}{\sum_{z^- \in Q^u} \exp(\text{sim}(f_t, z^-)/\tau)} \tag{5}$$

where $K^+ = \sum_{z^+ \in Q^d} \mathbf{1}(y^+ = \hat{y}_t)$, is the number of neighbours positively matched with $\hat{y}_t$.

**Case 2: Reliable sample from $C_u$.** If a test sample is identified as a reliable sample from $C_u$, we use the following contrastive objective by selecting positives $z^+$ from $Q_u$ and negatives $z^-$ from $Q_d$:

$$\mathcal{L}_U = -\frac{1}{K} \sum_{z^+ \in Q_u} \log \frac{\exp\left(\text{sim}\left(f_t, z^+\right)/\tau\right)}{\sum_{z^- \in Q_d} \exp(\text{sim}(f_t, z^-)/\tau)} \tag{6}$$

The LayerNorm parameters of the Vision Encoder are updated to minimize the following test time objective to adapt the model one sample at a time in an online manner:

$$\mathcal{L}_{ReDUCe} = \begin{cases} \mathcal{L}_{Re} + \mathcal{L}_D & \text{if} \quad s_t > \mu_d \\ \mathcal{L}_U & \text{if} \quad s_t < \mu_u \end{cases} \tag{7}$$

This objective improves the proximity between the test sample and its positives, suitably chosen based on its score $s_t$, while also pushing apart the test sample and its negatives. This collectively encourages the model to adapt such that each of the desired classes and undesired classes are clustered and farther apart from each other, improving the overall classification performance of $C_d$ and $C_u$. We now perform Gradient Analysis on the loss function and theoretically justify how the proposed ReDUCe loss helps in enhancing the discriminability between desired and undesired class samples.

**Evaluation Metrics.** We employ standard metrics, namely Area Under the Receiver Operating Characteristic Curve (AUROC) and False Positive Rate at a True Positive Rate of 95% (FPR95), from the OOD detection literature Lee et al. (2023); Li et al. (2023); Wang et al. (2023). Additionally, we compute the classification accuracy for desired class samples ($Acc_D$) and the binary classification accuracy for correctly recognizing samples from $C_u$ ($Acc_U$) as defined below. To gauge the overall performance, we compute $Acc_{HM}$ (HM), representing the harmonic mean of $Acc_D$ and $Acc_U$, which serves as a comprehensive metric capturing the trade-off between $Acc_D$ and $Acc_U$. Here, we

Table 1: Results with ImageNet-C/R as desired class data $D_d$, MNIST and SVHN for $D_u$.

| | Method | IN-C/MNIST | | | IN-C/SVHN | | | IN-R/MNIST | | | IN-R/SVHN | | |
|---|---|---|---|---|---|---|---|---|---|---|---|---|---|
| | | AUC ↑ | FPR ↓ | HM ↑ | AUC ↑ | FPR ↓ | HM ↑ | AUC ↑ | FPR ↓ | HM ↑ | AUC ↑ | FPR ↓ | HM ↑ |
| CLIP | ZS-Eval | 93.39 | 55.52 | 41.43 | 85.89 | 72.91 | 40.83 | 91.27 | 91.09 | 71.50 | 90.43 | 75.04 | 71.66 |
| | TPT | 93.12 | 58.01 | 42.21 | 85.43 | 74.47 | 40.95 | 91.25 | 91.23 | 71.98 | 90.43 | 74.98 | 72.36 |
| | TPT-C | 56.57 | 99.12 | 6.19 | 11.38 | 100.00 | 7.24 | 82.81 | 85.79 | 68.25 | 80.94 | 80.03 | 69.18 |
| | (K+1) PC | 95.76 | 10.43 | 42.95 | 87.75 | 26.23 | 38.50 | 97.46 | 11.78 | 81.51 | 97.55 | 11.17 | 80.39 |
| | TDA | 90.54 | 76.23 | 43.66 | 86.76 | 75.45 | 43.07 | 91.79 | 87.83 | 71.56 | 90.67 | 75.41 | 71.48 |
| | UniEnt | 94.19 | 46.98 | 41.53 | 87.56 | 67.03 | 41.10 | 91.64 | 88.67 | 71.73 | | | |
| | DPE | 87.92 | 91.94 | 42.87 | 82.96 | 77.90 | 41.93 | 92.13 | 81.09 | 71.39 | 90.86 | 73.30 | 70.64 |
| | ROSITA | **99.52** | **4.06** | **48.53** | **98.34** | **10.21** | **46.32** | **99.44** | **4.29** | **83.53** | **98.62** | **9.08** | **80.75** |
| | | +6.13 | +51.46 | +7.10 | +12.45 | +62.70 | +5.49 | +8.17 | +86.80 | +12.03 | +8.19 | +65.96 | +9.09 |
| MAPLE | ZS-Eval | 81.49 | 92.95 | 41.70 | 83.26 | 71.15 | 42.77 | 90.15 | 83.54 | 74.42 | 92.74 | 65.70 | 75.71 |
| | TPT | 81.38 | 93.17 | 39.92 | 83.18 | 71.52 | 40.93 | 90.14 | 83.58 | 74.00 | 92.74 | 65.68 | 75.23 |
| | TPT-C | 83.25 | 87.60 | 42.81 | 83.18 | 70.60 | 42.86 | 90.35 | 81.49 | 74.73 | 92.79 | 65.20 | 75.59 |
| | PAlign | 81.38 | 93.17 | 41.32 | 83.18 | 71.52 | 42.30 | 90.14 | 83.58 | 74.66 | 92.74 | 65.68 | 75.93 |
| | PAlign-C | 71.22 | 86.32 | 27.14 | 32.17 | 94.32 | 15.44 | 92.20 | 59.70 | 75.23 | 93.54 | 54.59 | 75.67 |
| | (K+1)PC | 98.58 | 3.35 | 48.69 | 77.17 | 39.74 | 38.10 | 99.01 | 3.16 | 84.23 | 95.14 | 13.77 | 80.16 |
| | TDA | 76.79 | 99.02 | 42.98 | 82.46 | 91.75 | 44.63 | 90.43 | 86.56 | 73.66 | 92.92 | 64.63 | 74.16 |
| | UniEnt | 81.53 | 93.45 | 41.50 | 83.41 | 70.84 | 42.78 | 90.14 | 83.49 | 74.48 | | | |
| | DPE | 73.97 | 99.59 | 41.39 | 80.06 | 87.10 | 44.05 | 90.44 | 78.77 | 72.67 | 93.48 | 55.74 | 76.74 |
| | ROSITA | **99.56** | **1.66** | **51.30** | **98.68** | **5.09** | **50.67** | **99.39** | **2.95** | **84.70** | **97.85** | **12.98** | **83.07** |
| | | +18.07 | +91.29 | +9.60 | +15.42 | +66.06 | +7.90 | +9.24 | +80.59 | +10.28 | +5.11 | +52.72 | +7.36 |

summarily report AUROC (AUC), FPR95 (FPR) and $Acc_{HM}$ (HM) for all the datasets (All five metrics are reported in detail in Appendix E).

$$Acc_D = \frac{\sum_{(x_i, y_i) \in \mathcal{D}_d} \mathbf{1}\left(y_i = \hat{y}_i\right) \cdot \mathbf{1}\left(y_i \in C_d\right)}{\sum_{(x_i, y_i) \in \mathcal{D}_d} \mathbf{1}\left(y_i \in C_d\right)}; \quad Acc_U = \frac{\sum_{(x_i, y_i) \in \mathcal{D}_u} \mathbf{1}\left(\hat{y}_i \in C_u\right) \cdot \mathbf{1}\left(y_i \in C_u\right)}{\sum_{(x_i, y_i) \in \mathcal{D}_u} \mathbf{1}\left(y_i \in C_u\right)}$$

### 3.1 GRADIENT ANALYSIS OF THE PROPOSED REDUCE LOSS

The key to understanding the behavior of the contrastive loss is to analyze its gradient. The softmax term in the denominator encourages $f_t$ to have lower similarity with negative samples, and the numerator encourages $f_t$ to have higher similarity with positive samples. We compute the gradient of the loss components $L_D$ and $L_U$ of the ReDUCe loss with respect to $f_t$ (Appendix A).

$$\frac{\partial \mathcal{L}_D}{\partial f_t} = -\frac{1}{K^+} \sum_{z^+ \in Q^d} \mathbf{1}\left(y^+ = \hat{y}_t\right) \cdot \frac{1}{\tau} \left(z^+ - \sum_{z^- \in Q^u} p\left(z^-\right) z^-\right)$$

$$\frac{\partial \mathcal{L}_U}{\partial f_t} = -\frac{1}{K} \sum_{z^+ \in Q^u} \frac{1}{\tau} \left(z^+ - \sum_{z^- \in Q^d} p\left(z^-\right) z^-\right)$$

(8)

where $p\left(z^-\right)$ is the softmax probability of the negative samples defined as

$$p\left(z^-\right) = \frac{\exp\left(\text{sim}\left(f_t, z^-\right)/\tau\right)}{\sum_{z' \in Q^-} \exp\left(\text{sim}\left(f_t, z'\right)/\tau\right)}$$

(9)

where $Q^-$ is $Q^u$ for $\mathcal{L}_D$ and $Q^d$ for $\mathcal{L}_U$. The gradient of these contrastive loss formulations drives the following behavior in this context:

1. **Attraction to positive neighbors.** In the gradient of $\mathcal{L}_D$, the first term pulls the test feature $f_t$ towards its positives $z^+ \in Q^d$, representing the attraction force that encourages samples from desired classes to form $|C_d|$ tight clusters as the positives are chosen such that $\hat{y}_t = y^+$. Similarly, in the gradient of $\mathcal{L}_U$, the first term pulls $f_t$ towards its positives $z^+ \in Q^u$ encouraging all samples from $C_u$ to cluster together.

2. **Repulsion from negative neighbors.** The second term $p\left(z^-\right) z^-$ in the gradient pushes the test feature $f_t$ away from its negatives $z^- \in Q^-$ ($Q^-$ is $Q^u$ for $\mathcal{L}_D$ and $Q^d$ for $\mathcal{L}_U$). The strength of the repulsion is controlled by the softmax probability $p(z^-)$, where more similar negatives exert a stronger repulsive force on $f_t$, increasing the separation between samples from $C_d$ and $C_u$. As the negatives selected are its $K$ nearest neighbours of the opposite type, they are infact hard negatives. Further, the contrastive objective inherently models the degree of hardness through the means of this probability $p(z^-)$. Closer the hard negative, stronger the repulsion force.

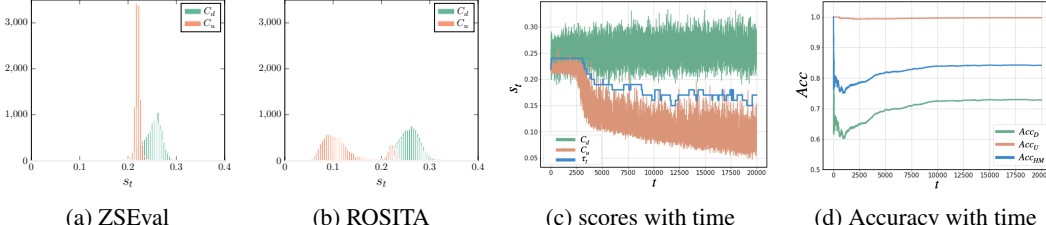

(a) ZSEval  (b) ROSITA  (c) scores with time  (d) Accuracy with time

Figure 3: Histograms of the scores $s_t$ for ZS-Eval (a) and ROSITA (b) on CIFAR-10C/MNIST dataset. (c) Change in scores for $C_d$ and $C_u$ class samples, the best threshold with time $t$; (d) Accuracy metrics measured for samples seen until time t. Using the LDA based class identifier with ROSITA, samples from $C_d$ and $C_u$ separate them better and the accuracy metrics improve with time.

**Key factors distinguishing ROSITA from prior works.**

1. *Enhanced Use of LDA Statistics to identify Reliable samples*: Apart from the threshold $\tau_t$, ROSITA leverages the score statistics $\mu_d$ and $\mu_u$ provided by the LDA class identifier, combined with the novel ReDUCe loss function, to adapt the model. This synergy enhances the discriminability between desired ($C_d$) and undesired ($C_u$) class samples, offering a clear advantage over baselines that use the same LDA identifier but fail to exploit this additional information (Figure 3).

2. *Bridging CNN and VLM-Based TTA Insights*: ROSITA integrates key insights from CNN-based TTA methods such as normalization layer updates with vision-language models (VLMs) (Section 2.3). While simple in hindsight, this baseline was overlooked in prior VLM-based TTA works Shu et al. (2022); Karmanov et al. (2024); Zhang et al. (2024). ROSITA highlights how these learnings can translate effectively to VLMs, underscoring their utility as a foundational approach for TTA.

3. *Holistic Design for Open-set TTA*: ROSITA introduces the ReDUCe loss to distinctly separate desired ($C_d$)and undesired ($C_u$) class samples using compact feature banks. Although it is inspired by contrastive learning frameworks Chen et al. (2020; 2022), it is specifically designed for open-set TTA: (i) Reliable samples from $C_u$ use nearest $C_u$ samples as negatives, and vice versa (ii) Unlike the $C_d$+1-way classification in Li et al. (2023), ROSITA forces $C_d$ features to form distinct clusters and pushes $C_u$features away. (iii) The feature banks are populated only with reliable samples, ensuring robust updates during adaptation (see Appendix C.5). This approach addresses the significant overlap of zero-shot scores $s_t$ between $C_d$ and $C_u$ in vision-language models, reducing misclassification and boosting discriminability.

## 4 EXPERIMENTS

**Datasets.** We experiment with a diverse set of datasets to choose desired class data $D_d$ and undesired class data $D_u$. For $D_d$, we use CIFAR-10C Hendrycks & Dietterich (2019), CIFAR-100C Hendrycks & Dietterich (2019), ImageNet-C Hendrycks & Dietterich (2019) from the corruption category and ImageNet-R Hendrycks et al. (2021), VisDA Peng et al. (2017) and the Clipart, Painting, Sketch domains from DomainNet Peng et al. (2019a) as style transfer datasets. We introduce samples from MNIST Le-Cun et al. (1998), SVHN Netzer et al. (2011), CIFAR-10/100C Hendrycks & Dietterich (2019) and TinyImageNet Le & Yang (2015) datasets as $D_u$ in the test stream. We describe the datasets in detail in the Appendix B.3.

Table 2: $Acc_{HM}$ on VisDA dataset and Clipart, Painting, Sketch domains from DomainNet as $D_d$ and MNIST as $D_u$.

| Method | VisDA | Clipart | Painting | Sketch |
|---|---|---|---|---|
| ZSEval | 78.28 | 50.22 | 47.81 | 48.59 |
| TPT | 78.42 | 57.71 | 49.73 | 54.67 |
| TPT-C | 75.35 | 57.57 | 49.31 | 54.41 |
| (K+1)PC | 90.35 | 71.21 | 70.61 | 67.21 |
| TDA | 76.85 | 61.04 | 51.20 | 55.26 |
| UniEnt | 78.09 | 57.88 | 49.75 | 54.76 |
| DPE | 53.67 | 54.52 | 47.91 | 32.18 |
| ROSITA | 90.64 | 71.40 | 70.89 | 67.35 |
| | +12.36 | +21.18 | +23.08 | +18.76 |

**Implementation Details.** We use CLIP and MaPLe backbones with ViT-B16 architecture. For ROSITA, we use SGD optimizer with a learning rate of 0.001 to update the LayerNorm parameters of the Vision encoder. We set size of the score bank $\mathcal{S}$ to 512, number of neighbours $K$ to 5. The size of feature bank $M_d$ is set as $K \times C_d$ and that of $M_u$ to 64. Implementation details for all the baseline methods are presented in Appendix B.4 *We equip all methods with the same $C_d$ vs $C_u$ class identifier described in Section 2.4.* All experiments are done on a single NVIDIA A6000 GPU.

Table 3: Results with CIFAR-10C/100C as desired class data $D_d$ and four other datasets as $D_u$.

| | Method | MNIST | | | SVHN | | | Tiny-ImageNet | | | CIFAR-100C/10-C | | |
|---|---|---|---|---|---|---|---|---|---|---|---|---|---|
| | | AUC ↑ | FPR ↓ | HM ↑ | AUC ↑ | FPR ↓ | HM ↑ | AUC ↑ | FPR ↓ | HM ↑ | AUC ↑ | FPR ↓ | HM ↑ |
| **CIFAR-10C / CLIP** | ZS-Eval | 91.91 | 85.04 | 75.57 | 89.93 | 64.20 | 74.08 | 91.33 | 27.07 | 74.63 | 82.57 | 67.92 | 68.89 |
| | TPT | 91.89 | 85.55 | 75.81 | 89.93 | 64.41 | 74.36 | 91.31 | 27.23 | 75.17 | 82.57 | 68.06 | 69.17 |
| | TPT-C | 81.64 | 67.53 | 74.86 | 58.48 | 71.72 | 48.26 | 74.08 | 61.45 | 49.88 | 61.45 | 94.30 | 46.10 |
| | (K+1)PC | 98.05 | 12.50 | 83.27 | 80.74 | 50.33 | 70.10 | 87.09 | 52.29 | 73.98 | 62.55 | 91.68 | 56.46 |
| | UniEnt | 91.98 | 85.2 | 75.62 | 89.97 | 64.38 | 74.18 | 91.40 | 26.96 | 74.73 | 82.59 | 68.14 | 68.98 |
| | TDA | 92.94 | 71.11 | 77.06 | 92.02 | 52.68 | 76.64 | 91.68 | 25.37 | 75.94 | 83.54 | 66.06 | 70.13 |
| | DPE | 46.97 | 99.10 | 27.60 | 84.15 | 85.24 | 68.52 | 89.92 | 31.30 | 69.90 | 79.18 | 75.06 | 62.34 |
| | **ROSITA** | **99.10** | **7.63** | **84.17** | **94.79** | **32.59** | **78.80** | **96.43** | **12.10** | **80.06** | **82.99** | **62.89** | **69.56** |
| | | +7.19 | +77.41 | +8.60 | +4.86 | +31.61 | +4.72 | +5.10 | +14.97 | +5.43 | +0.42 | +5.03 | +0.6 |
| **CIFAR-10C / MAPLE** | ZS-Eval | 98.48 | 3.77 | 83.63 | 98.34 | **7.86** | 83.57 | 90.86 | 27.54 | 76.04 | 86.14 | 52.08 | 71.76 |
| | TPT | 98.15 | 5.67 | 81.56 | 98.34 | 7.89 | 82.73 | 90.86 | 27.61 | 75.46 | 86.15 | 52.14 | 70.94 |
| | TPT-C | 98.56 | **3.74** | 83.51 | 98.32 | 8.18 | 83.47 | 91.18 | 26.93 | 76.31 | 86.50 | 50.56 | 71.07 |
| | PAlign | 98.15 | 5.67 | 82.24 | **98.34** | 7.90 | 83.51 | 90.86 | 27.60 | 75.98 | 86.15 | 52.18 | 71.52 |
| | PAlign-C | 98.56 | 3.74 | 83.49 | 98.32 | 8.13 | 83.46 | 91.18 | 26.90 | 76.30 | 86.50 | 50.58 | 71.04 |
| | (K+1)PC | 98.34 | 9.63 | 86.52 | 71.01 | 78.78 | 68.70 | 71.20 | 85.81 | 68.29 | 62.35 | 88.44 | 61.89 |
| | UniEnt | 98.17 | 5.49 | 82.64 | 98.35 | 7.85 | 83.65 | 90.90 | 27.41 | 76.08 | 86.16 | 51.91 | 71.72 |
| | TDA | 98.42 | 4.13 | 81.97 | 98.60 | 6.20 | 83.95 | 91.27 | 27.00 | 76.84 | 86.72 | 51.40 | 72.61 |
| | DPE | 83.82 | 92.73 | 55.52 | 97.42 | 12.95 | 79.41 | 89.10 | 31.13 | 74.32 | 73.57 | 73.67 | 53.64 |
| | **ROSITA** | **99.34** | 5.22 | **87.63** | 97.80 | 13.15 | **84.17** | **91.67** | **25.31** | **77.67** | **86.82** | 50.33 | **73.15** |
| | | +0.86 | -1.45 | +4.00 | +0.54 | -5.29 | +0.60 | +0.81 | +2.23 | +1.63 | +0.68 | +1.75 | +1.39 |
| **CIFAR-100C / CLIP** | ZS-Eval | 77.78 | 99.93 | 48.39 | 64.70 | 98.68 | 45.85 | 67.31 | 73.89 | 45.80 | 63.28 | 93.25 | 44.04 |
| | TPT | 77.76 | 99.94 | 48.33 | 64.71 | 98.63 | 45.85 | 67.28 | 73.82 | 45.93 | 63.26 | 93.20 | 44.02 |
| | TPT-C | 51.57 | 100.00 | 27.04 | 9.40 | 99.98 | 5.74 | 59.74 | 79.76 | 18.41 | 55.86 | **86.35** | 13.64 |
| | (K+1)PC | 96.89 | 12.15 | 59.72 | 75.24 | 51.64 | 43.73 | 41.84 | 99.61 | 31.83 | 54.02 | 93.93 | 32.00 |
| | TDA | 80.33 | 99.57 | 46.52 | 71.77 | 96.11 | 46.01 | 70.70 | 69.63 | 47.52 | 66.07 | 91.90 | 45.79 |
| | UniEnt | 77.94 | 99.93 | 48.32 | 64.78 | 98.61 | 45.84 | 67.40 | 73.77 | 45.83 | 63.28 | 93.18 | 44.04 |
| | DPE | 67.06 | 99.88 | 42.54 | 43.23 | 99.79 | 35.69 | 61.42 | 80.62 | 42.80 | 60.08 | 92.80 | 42.21 |
| | **ROSITA** | **96.07** | **19.28** | 57.34 | **82.09** | **64.64** | **48.17** | **83.55** | **50.76** | **55.88** | **68.54** | 89.71 | **47.98** |
| | | +18.29 | +80.65 | +8.95 | +17.39 | +34.04 | +2.32 | +16.24 | +23.13 | +10.08 | +5.26 | -3.54 | +3.94 |
| **CIFAR-100C / MAPLE** | ZS-Eval | 87.43 | 64.19 | 54.97 | 92.98 | 40.51 | 56.42 | 68.80 | 74.35 | 48.24 | 66.93 | 87.94 | 46.06 |
| | TPT | 87.42 | 64.09 | 53.09 | 92.97 | 40.44 | 54.37 | 68.80 | **74.20** | 46.97 | 66.93 | 87.95 | 44.38 |
| | TPT-C | 87.65 | 63.08 | 55.14 | 93.09 | 40.30 | 56.31 | 68.85 | 74.71 | 48.53 | 66.97 | 87.94 | 46.30 |
| | PAlign | 87.42 | 64.11 | 53.98 | 92.97 | 40.48 | 55.37 | 68.80 | 74.23 | 47.69 | 66.93 | 87.93 | 45.16 |
| | PAlign-C | 88.25 | 57.31 | 55.69 | 93.45 | 39.39 | 57.39 | 68.76 | 78.12 | 48.15 | 66.82 | 87.80 | 47.01 |
| | (K+1)PC | 96.49 | 9.42 | 62.97 | 65.73 | 78.63 | 32.60 | 42.94 | 99.95 | 27.52 | 53.48 | 94.26 | 34.70 |
| | TDA | 89.82 | 52.24 | 55.46 | 95.04 | 30.76 | 59.51 | 72.05 | 71.83 | 49.19 | 69.12 | 87.36 | 49.06 |
| | UniEnt | 87.40 | 64.02 | 54.86 | 92.99 | 40.36 | 56.42 | 68.84 | 74.26 | 48.41 | 66.93 | 87.96 | 46.09 |
| | DPE | 39.05 | 98.88 | 33.66 | 84.29 | 76.13 | 52.20 | 63.74 | 82.75 | 45.74 | 65.61 | 90.67 | 46.36 |
| | **ROSITA** | **97.04** | **11.01** | 62.06 | **96.26** | **20.99** | 59.25 | 70.37 | 77.00 | 48.68 | **69.57** | **83.61** | 48.80 |
| | | +9.61 | +53.18 | +7.09 | +3.28 | +19.52 | +2.83 | +1.57 | -2.65 | +0.44 | +2.64 | +4.33 | +2.74 |

## 5 ANALYSIS

**Comparison with prior methods.** We observe, from Table 1, 2, 3 that TPT and PAlign perform similar to ZSEval in most datasets, as the prompts are reset after every single image update. On continuously updating prompts in TPT-C and PAlign-C, we observe a reduction in HM compared to ZS-Eval. The effect is more severe with CLIP when compared to MaPLe, as only the text prompts are updated keeping the vision encoder fixed (as also observed in Section 2.3). **ROSITA**, being equipped with a carefully designed objective to better discriminate between samples from $C_d$ and $C_u$ samples (Figure 3), results in overall better metrics in general. *We study the need for reliable samples in C.5, analyse the sensitivity of ROSITA's performance for different random seeds in C.1, choice of parameter K in C.2. We report additional experimental results using CLIP with ViT-B/32 and ResNet-50 architecture in D.2 and with different corruption types in D.1.*

**Performance in different Open set TTA scenarios. (a) Continuously changing domains:** We sequentially present 15 corruptions from CIFAR-10C, which form the domain $D_d$, alongside samples from four other datasets $D_u$. **(b) Frequently changing domains:** To further simulate more dynamic test environments, for CIFAR-10C/MNIST, we reduce the number of samples per corruption to 100, 250, 500, and 1000 in the continuously changing domain open-set TTA scenario. Reducing the sample count per corruption causes more frequent domain changes, increasing the challenge for adaptation. **(c) Varying ratio of samples belonging to classes $C_d$ vs $C_u$:** We simulate real-world scenarios using the CIFAR-10C/MNIST dataset by varying the ratio of samples from the known classes $C_d$ versus unknown classes $C_u$ in the test stream by varying this ratio as 0.2, 0.4, 0.6, and 0.8. From results in Table 5,we observe that **ROSITA** demonstrates consistent superiority across all

Table 5: Performance in different Open set TTA scenarios.

| Method | (a) Continuously changing domains | | | | (b) Frequently changing domains | | | | (c) Varying ratio of $C_d/C_u$ | | | |
|---|---|---|---|---|---|---|---|---|---|---|---|---|
| | CIFAR-10C | | | | No. of samples per corruption | | | | Ratio | | | |
| | SVHN | MNIST | Tiny | C-100C | 100 | 200 | 500 | 1000 | 0.2 | 0.4 | 0.6 | 0.8 |
| ZSEval | 64.33 | 64.04 | 66.50 | 58.49 | 61.41 | 61.87 | 61.42 | 63.30 | 75.56 | 75.59 | 75.57 | 75.56 |
| TPT | 64.26 | 64.03 | 66.50 | 58.47 | 61.33 | 62.32 | 61.59 | 63.24 | 75.67 | 75.75 | 75.81 | 75.83 |
| TPT-C | 33.05 | 46.44 | 59.38 | 37.24 | 60.62 | 61.30 | 57.16 | 34.88 | 72.70 | 74.31 | 74.79 | 75.16 |
| (K+1)PC | 65.13 | 62.52 | 66.93 | 57.46 | 60.90 | 60.76 | 61.40 | 63.26 | 62.31 | 68.85 | 81.70 | 82.90 |
| TDA | 66.02 | 66.44 | 67.64 | 59.44 | 60.17 | 61.43 | 63.22 | 64.82 | 72.45 | 75.04 | 77.54 | 77.91 |
| DPE | 23.36 | 50.12 | 58.96 | 35.56 | 47.48 | 46.22 | 39.83 | 46.52 | 65.67 | 66.12 | 56.38 | 29.98 |
| ROSITA | **66.86** | **65.26** | **68.89** | **59.16** | **61.64** | **66.82** | **67.97** | **73.24** | **82.96** | **83.97** | **84.51** | **84.37** |

three open-set TTA scenarios, showcasing its capability to adapt effectively to both continuously and frequently changing domains, as well as varying class distributions.

**Loss Ablation.** We observe that only using $\mathcal{L}_{Re}$ or $\mathcal{L}_D$ improves the metrics for CIFAR-10C dataset. For ImageNet-R (IN-R) as $D_d$, using $\mathcal{L}_{Re}$ or $\mathcal{L}_D$ is observed to increase FPR and decrease HM. IN-R has 200 classes making it a more challenging and confusing task compared to CIFAR-10C. This decrease in performance for IN-R can be attributed to the misclassification of some samples from $C_u$ as reliable desired class samples, increasing the confusion between

Table 4: Ablation study on loss components.

| $\mathcal{L}_{Re}$ | $\mathcal{L}_D$ | $\mathcal{L}_U$ | CIFAR-10C/MNIST | | | IN-R/MNIST | | |
|---|---|---|---|---|---|---|---|---|
| | | | AUC ↑ | FPR ↓ | HM ↑ | AUC ↑ | FPR ↓ | HM ↑ |
| ✗ | ✗ | ✗ | 91.91 | 85.04 | 75.57 | 91.27 | 91.09 | 71.5 |
| ✓ | ✗ | ✗ | 95.29 | 30.82 | 80.97 | 81.07 | 99.02 | 64.32 |
| ✗ | ✓ | ✗ | 95.23 | 28.91 | 79.71 | 87.73 | 94.67 | 67.28 |
| ✗ | ✗ | ✓ | 98.61 | 12.73 | 79.84 | 99.39 | 4.81 | 80.82 |
| ✓ | ✓ | ✗ | 96.23 | 22.73 | 79.24 | 76.78 | 99.22 | 62.54 |
| ✓ | ✗ | ✓ | 98.69 | 12.06 | 82.98 | 99.34 | 4.67 | 82.98 |
| ✗ | ✓ | ✓ | **99.27** | **4.15** | 80.69 | 99.48 | 4.40 | 81.92 |
| ✓ | ✓ | ✓ | 99.10 | 7.63 | **84.17** | 99.44 | 4.29 | **83.53** |

$C_d$ and $C_u$ classes. Using $\mathcal{L}_U$ significantly reduces the confusion between samples from $C_d$ and $C_u$, shown by the significant drop in FPR compared to ZSEval. The contrastive objectives $\mathcal{L}_D$ and $\mathcal{L}_U$ to separate the two types of samples, in conjunction with reliable pseudo label loss $\mathcal{L}_{Re}$ which aids to improve the $|C_d|$-way classification of desired class samples, gives the overall best results.

**Memory buffer.** Prior prompt tuning methods like TPT Shu et al. (2022), Samadh et al. (2023) do not require any memory buffer. TDA Karmanov et al. (2024) requires a memory buffer of size $(|C_d| \times (3 + 2)) \times F$ to store 3 features per desired class in the positive cache and 2 features per class in the negative cache. DPE Zhang et al.

Table 6: Memory overhead in ROSITA.

| Dataset | $|C_d|$ | No. of features | Memory (in MB) |
|---|---|---|---|
| CIFAR-10C | 10 | 5x10+64 | 0.758 |
| VisDA | 12 | 5x12+64 | 0.778 |
| CIFAR-100C | 100 | 5x100+64 | 1.679 |
| ImageNet-R | 200 | 5x200+64 | 2.703 |
| ImageNet-C | 1000 | 5x1000+64 | 10.89 |

(2024) requires a memory buffer of size $(|C_d| \times 3) \times F$ to store 3 features per desired class. ROSITA requires a small memory buffer of size 512 for the score bank $S$ and $(|C_d| \times K + |M_u|) \times F$ for the feature banks. For a ViT-B16 ($F = 512$) model with ImageNet-C ($|C_d| = 1000$), the required memory buffer size is $5 \times 1000 \times 512 + 64 \times 512$ (10.89MB). *The memory to store them and computation required to compute feature similarity is as lightweight as performing a forward pass through a simple linear layer, demonstrating the memory and computational efficiency of* **ROSITA** *for real time applications.*

**Complexity Analysis** For prompt tuning methods TPT/-C and PAlign/-C, the GPU memory and time taken (secs/image) scales with the number of classes, as it requires more memory to store the intermediate activations and gradients. The time taken to perform forward and backward pass through the text encoder also depends on the number of classes. On the other hand, ROSITA requires two forward passes and one backward pass through the vision encoder for reliable test samples. For e.g., for ImageNet-C dataset with 1000 classes, ZSEval, TPT, TDA and ROSITA require 5.71 GB, 23.24 GB, 5.71 GB and 5.73 GB GPU memory (refer Appendix C.8) to perform a single image based model update. Hence, **ROSITA** is computationally very efficient, similar to that of ZSEval.

# 6 CONCLUSION

In this work, we propose **ROSITA**, a novel framework to address the challenging problem Open set Test Time Adaptation (TTA) on a single image basis. ROSITA effectively distinguishes between samples from desired classes vs others by leveraging two dynamically updated feature banks. The proposed ReDUCe loss facilitates effective model adaptation by using reliable, while mitigating any negative impact of undesirable samples in the test stream. Through extensive experimentation on diverse domain adaptation benchmarks, we demonstrate the effectiveness of ROSITA in several scenarios inspired by the dynamic real world environment. We discuss the limitations in B.5.

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

# APPENDIX

## A    GRADIENT ANALYSIS OF THE REDUCE LOSS

Here, we delve deeper into the ReDUCe loss function in ROSITA, breaking down its key components and mathematically demonstrate why the proposed objective improves the separation of $C_d$ and $C_u$ samples. We'll focus on contrastive loss components $L_D$ and $L_U$ which are designed to improve discriminability.

**ReDUCe loss in a nutshell.** A test sample $x_t$ arrives at time $t$ with feature representation $f_t$. Two feature banks, $\mathcal{M}_w$ and $\mathcal{M}_s$ store reliable sample features from $C_d$ and $C_u$ respectively. ReDUCe loss aims to pull the test sample's feature $f_t$ towards its positive samples $z^+$, which are its K nearest neighbors $Q^d = \text{k}NN(f_t; M_d)$ if it is a reliable $C_d$ sample or $Q^u = \text{k}NN(f_t; M_u)$ if it is a reliable $C_u$ sample. The feature $f_t$ is pushed away from its negative samples $z^-$, which are the K nearest neighbors from the undesired feature bank $M_u$ if it is a reliable $C_d$ sample or from the desired feature bank $M_d$ if it is a reliable $C_u$ sample. The features $f_t, z^+, z^-$ are all unit norm vectors.

The key to understanding the behavior of the contrastive loss is to analyze its gradient. **Gradient of $L_D$ with respect to $f_t$:**

The contrastive loss for desired class samples $L_D$ is defined as:

$$\mathcal{L}_D = -\frac{1}{K^+} \sum_{z^+ \in Q^d} \mathbf{1}(y^+ = \hat{y}_t) \log \frac{\exp\left(\text{sim}\left(f_t, z^+\right)/\tau\right)}{\sum_{z^- \in Q^u} \exp(\text{sim}(f_t, z^-)/\tau)}$$

$$\frac{\partial \mathcal{L}_D}{\partial f_t} = -\frac{1}{K^+} \sum_{z^+ \in Q^d} \mathbf{1}(y^+ = \hat{y}_t) \frac{\partial}{\partial f_t} \log \frac{\exp\left(\text{sim}\left(f_t, z^+\right)/\tau\right)}{\sum_{z^- \in Q^u} \exp(\text{sim}(f_t, z^-)/\tau)}$$

$$(10)$$

The loss is of the log-softmax structure. Consider gradient of the following term:

$$\frac{\partial}{\partial f_t} \log \frac{\exp\left(\text{sim}\left(f_t, z^+\right)/\tau\right)}{\sum_{z^- \in Q} \exp(\text{sim}(f_t, z^-)/\tau)} = \frac{\partial}{\partial f_t}\left(\frac{\text{sim}\left(f_t, z^+\right)}{\tau}\right) - \frac{\partial}{\partial f_t} \log \sum_{z^- \in Q} \exp(\text{sim}(f_t, z^-)/\tau)$$

The gradients of the two terms involved are

$$\frac{\partial}{\partial f_t}\left(\frac{\text{sim}\left(f_t, z^+\right)}{\tau}\right) = \frac{z^+}{\tau}$$

$$\frac{\partial}{\partial f_t} \log \sum_{z^- \in Q} \exp(\text{sim}(f_t, z^-)/\tau) = \frac{\sum_{z^- \in Q} \frac{\partial}{\partial f_t} \exp(\text{sim}(f_t, z^-)/\tau)}{\sum_{z^- \in Q} \exp(\text{sim}(f_t, z^-)/\tau)}$$

$$= \frac{1}{\tau} \cdot \frac{\sum_{z^- \in Q} \exp(\text{sim}(f_t, z^-)/\tau)}{\sum_{z^- \in Q} \exp(\text{sim}(f_t, z^-)/\tau)z^-}$$

$$= \frac{1}{\tau} \cdot \sum_{z^- \in Q} p(z^-)z^-$$

The final gradient of the log-softmax term is

$$\frac{\partial}{\partial f_t} \log \frac{\exp\left(\text{sim}\left(f_t, z^+\right)/\tau\right)}{\sum_{z^- \in Q} \exp(\text{sim}(f_t, z^-)/\tau)} = \left(z^+ - \sum_{z^- \in Q} p\left(z^-\right) z^-\right)$$

$$(11)$$

where $p\left(z^-\right)$ is the softmax probability of the negative samples defined as

$$p\left(z^{-}\right) = \frac{\exp\left(\text{sim}\left(f_t, z^{-}\right)/\tau\right)}{\sum\limits_{z' \in Q^{-}} \exp\left(\text{sim}\left(f_t, z'\right)/\tau\right)}$$

Substituting Equation 11 in Equation 10, we get the gradient of the desired sample contrastive loss $L_D$ with respect to $f_t$ as

$$\frac{\partial \mathcal{L}_D}{\partial f_t} = -\frac{1}{K^+} \sum_{z^+ \in Q^d} \mathbf{1}(y^+ = \hat{y}_t) \left(z^+ - \sum_{z^- \in Q^u} p\left(z^-\right) z^-\right) \tag{12}$$

**Gradient of $L_D$ with respect to $f_t$:**

The contrastive loss for desired class samples $L_D$ is defined as:

$$\mathcal{L}_U = -\frac{1}{K} \sum_{z^+ \in Q^u} \log \frac{\exp\left(\text{sim}\left(f_t, z^+\right)/\tau\right)}{\sum\limits_{z^- \in Q^d} \exp(\text{sim}(f_t, z^-)/\tau)}$$

$$\frac{\partial \mathcal{L}_U}{\partial f_t} = -\frac{1}{K^+} \sum_{z^+ \in Q^u} \frac{\partial}{\partial f_t} \log \frac{\exp\left(\text{sim}\left(f_t, z^+\right)/\tau\right)}{\sum\limits_{z^- \in Q^d} \exp(\text{sim}(f_t, z^-)/\tau)} \tag{13}$$

Substituting Equation 11 in Equation 13, we get:

$$\frac{\partial \mathcal{L}_U}{\partial f_t} = -\frac{1}{K^+} \sum_{z^+ \in Q^u} \left(z^+ - \sum_{z^- \in Q^d} p\left(z^-\right) z^-\right) \tag{14}$$

**Interpretation of the Gradients.**

- Both the gradient terms in Equations 12 and 14 have two components: Positive term $z^+$ and Negative term $p\left(z^-\right) z^-$. The positives and negatives are suitably chosen from the desired and undesired feature banks.

- Positive term $z^+$: The term $z^+$ pulls the test feature $f_t$ closer to its feature vectors $z^+$. This term represents the attraction force that encourages $C_d$ samples to cluster together in $L_D$ and $C_u$ samples to cluster together in $L_U$.

- Negative term $p\left(z^-\right) z^-$: The negative samples $z^-$ exert a repulsive force, pushing $f_t$ away from them. The strength of this repulsion is controlled by the softmax probabilities $p\left(z^-\right)$, where higher similarity between $f_t$ and $z^-$ increases the repulsion force. This inherently models the degree of hard negatives from the negative feature bank.

- The overall gradient update encourages $f_t$ to move closer to its positives while moving away from its negatives, enhancing the separation between samples from $C_d$ and $C_u$ classes.

# B  BASELINES

## B.1  VISION LANGUAGE MODELS

**CLIP** Radford et al. (2021) is a multimodal VLM consisting of two modules: Vision encoder and Text encoder denoted as $\mathcal{F}_V$ and $\mathcal{F}_T$ respectively. During pre-training, the two modules are jointly trained in a contrastive self-supervised fashion to align massive amounts of web scrapped image-text pairs. CLIP has demonstrated impressive zero-shot performance across a wide variety of datasets.

**MaPLe** Khattak et al. (2023) is a multimodal prompt learner model that simultaneously adapts both the vision and text encoders while finetuning CLIP for downstream tasks. They use learnable text prompts $\boldsymbol{p}_T$ and bridge the two modalities using visual prompts obtained as $\boldsymbol{p}_V = \text{Proj}(\boldsymbol{p}_T)$. Learnable tokens are also introduced in the deeper layers of both image and text encoders, to enable progressive adaptation of the features.

## B.2  METHODS

**ZSEval (Radford et al., 2021):** Given a test image $x_t$, the image feature is extracted from the vision encoder as $f_t = \mathcal{F}_V(x_t)$. For a $C$-class classification problem, the classifier is obtained by prepending a predefined text prompt $\boldsymbol{p}_T$="A photo of a", with the class names $\{c_1, c_2, \ldots c_C\}$ to form class specific text inputs $\{\boldsymbol{p}_T, c_i\}$ for $i \in \{1, \ldots C\}$. These texts are then embedded through the text encoder as $\boldsymbol{t}_i = \mathcal{F}_T(\{\boldsymbol{p}_T; c_i\})$ to get the text classifiers $\{\boldsymbol{t}_1, \boldsymbol{t}_2, \ldots \boldsymbol{t}_C\}$. The class prediction is made by identifying the text feature $\boldsymbol{t}_i$ which has the highest similarity with the image feature $f_t$.

**TPT Shu et al. (2022)** aims to improve the zero shot generalization ability of CLIP by providing custom adaptable context for each image. This is done by prepending learnable text prompts $\boldsymbol{p}_T$ to the class names instead of a predefined text prompt. The text classifiers $\boldsymbol{t}_i = \mathcal{F}_T(\{\boldsymbol{p}_T; c_i\}), i \in \{1, 2, \ldots C\}$ are now a function of these learnable prompts, which are specially adapted for each test image using an entropy minimization objective as $\arg\min_{\boldsymbol{p}_T} \mathcal{L}_{\text{ent}}$. The entropy is obtained using the average score vector of the filtered augmented views.

**PromptAlign (PAlign) (Samadh et al., 2023)** leverages multimodal prompt learner model MaPLe Khattak et al. (2023) to facilitate the adaptation of both vision and language encoders for each test sample. They align the token distributions of source and target domains, considering ImageNet as a proxy for the source dataset of CLIP. The vision and language prompts of MaPLe are optimized with the objective $\arg\min_{\{\boldsymbol{p}_V, \boldsymbol{p}_T\}} \mathcal{L}_{ent} + \mathcal{L}_{align}$ for each sample $x_t$.

**TPT-C Shu et al. (2022)/PAlign-C (Samadh et al., 2023)**: We adapt TPT and PAlign for continuous model update, which we refer as TPT-C and PAlign-C respectively. The prompts $\{\boldsymbol{p}_T\}$ and $\{\boldsymbol{p}_V, \boldsymbol{p}_T\}$ in TPT and PAlign are continuously updated with the test stream with their respective test objectives.

**(K+1)PC (Li et al., 2023)**: This was the first work exploring open world TTA, however it was done in the context of CNNs and not VLMs. Also, the test samples come in batches, while we perform single image TTA. We adapt this method for our problem setting as follows: As we use VLMs, we use the text prototypes (instead of the source prototypes). The prototype pool is dynamically updated by adding features of reliable test samples recognized to belong to undesired classes. The vision encoder is updated using a (K+1) way prototypical cross entropy loss.

**TDA (Karmanov et al., 2024)**: TDA is a training-free dynamic adapter for test-time adaptation in vision-language models, utilizing a lightweight key-value cache for efficient pseudo label refinement without backpropagation.

**DPE (Zhang et al., 2024)**: DPE accumulates task-specific knowledge by dynamically evolving two sets of prototypes, textual and visual, during test time. These prototypes are refined to capture increasingly accurate multi-modal representations for target classes. To ensure consistency between modalities, DPE incorporates learnable residuals for each test sample, aligning textual and visual prototypes for improved representation alignment.

**UniEnt Gao et al. (2024)**: This is a very recent work addressing open-set TTA in the context of CNNs. They use a Distribution Aware Filter (DAF) based on Gaussian Mixture Modeling of the scores to distinguish between desired and undesired class samples. They employ entropy minimization and entropy maximization objectives for desired and undesired class samples respectively.

We equip all the baselines with the same LDA based desired vs undesired class identifier described in Section 2.4 for fair comparison of the TTA methods for this problem.

### B.3 DATASETS

We experiment with a diverse set of datasets, encompassing corruption datasets, style transfer datasets, and other common datasets.

**CIFAR10-C** Hendrycks & Dietterich (2019) is a small-scale corruption dataset of 10 classes with 15 common corruption types. It consists of 10,000 images for each corruption.

**CIFAR-100C** Hendrycks & Dietterich (2019) is also a corruption dataset with 100 classes and 15 corruption types. It also consists of 10,000 images for each corruption.

**ImageNet-C** Hendrycks & Dietterich (2019) is a large-scale corruption dataset spanning 1000 categories with a total of 50,000 images. 15 types of corruption images are synthesized from these 50,000 images.

**ImageNet-R** Hendrycks et al. (2021) is a realistic style transfer dataset encompassing interpretations of 200 ImageNet classes, amounting to a total of 30,000 images.

**VisDA** Peng et al. (2017) is a synthetic-to-real large-scale dataset, comprising of 152,397 synthetic training images and 55,388 real testing images across 12 categories.

**DomainNet** Peng et al. (2019a) is a large-scale domain adaptation dataset. We use the Clipart, Painting and Sketch domains with 345 categories from the DomainNet dataset for our experiments.

**MNIST** LeCun et al. (1998) is a dataset of handwritten images consisting of 60,000 training and 10,000 testing images.

**SVHN** Netzer et al. (2011) is also a digits dataset with house numbers captured from real streets. It consists of 50,000 training images and 10,000 testing images.

We perform experiments on eight domains $D_d$ for desired class samples. The corresponding $D_u$ are chosen such that there is no overlap between the classes $C_d$ and $C_u$ as described in Table 7. The 15 corruptions of CIFAR-10C/100C and ImageNet-C fall into four categories: synthetic weather effects, per-pixel noise, blurring, and digital transforms. *snow* corruption is a synthesized weather effect on which all the main experiments of CIFAR-10C, CIFAR-100C and ImageNet-C are done. To evaluate the robustness of our method across different corruption types, we do additional experiments with *impulse noise* , *motion blur* and *jpeg compression* corruptions from the categories per-pixel noise, blurring and digital transforms respectively and report the results in Section D.1.

Table 7: Details of desired and undesired class dataset combinations

| Datasets | | # images | | |
| --- | --- | --- | --- | --- |
| $D_d$ | $D_u$ | $D_d$ | $D_u$ | Total |
| CIFAR-10C | MNIST, SVHN, Tiny ImageNet, CIFAR-100C | 10000 | 10000 | 20000 |
| CIFAR-100C | MNIST, SVHN, Tiny ImageNet, CIFAR-10C | 10000 | 10000 | 20000 |
| ImageNet-C | MNIST, SVHN | 50000 | 50000 | 100000 |
| ImageNet-R | MNIST, SVHN | 30000 | 30000 | 60000 |
| VisDA | MNIST, SVHN | 50000 | 50000 | 100000 |
| Clipart | MNIST, SVHN | 29208 | 29208 | 58416 |
| Painting | MNIST, SVHN | 43700 | 43700 | 87400 |
| Sketch | MNIST, SVHN | 41832 | 41832 | 83664 |

### B.4 IMPLEMENTATION DETAILS

Here, we describe the parameters chosen for all the baseline methods and our proposed method.

**TPT Shu et al. (2022):** The prompt is initialized with the default *A photo of a* text. The corresponding 4 tokens in the input text embedding space are optimized for each test image. The prompt is **reset**

after each update. A single test image is augmented 63 times using random resized crops to create a batch of 64 images. The confident samples with 10% lowest entropy are selected. The test time loss is the entropy of the averaged prediction of the selected confident samples. AdamW optimizer with a learning rate of $5e^{-4}$ is used, following Shu et al. (2022).

**PAlign Samadh et al. (2023):** Following PromptAlign Samadh et al. (2023), MaPLe Khattak et al. (2023) model trained on ImageNet using 16-shot training data with 2 prompt tokens for a depth of 3 layers is used. The prompts on both the text and vision encoders are optimized on a single test image. Similar to TPT, 10% of 64 augmentations are selected to compute the entropy loss. The token distribution loss to align the token statistics of test with that of source data is computed for all 64 images. AdamW optimizer with a learning rate of $5e^{-4}$ to update the prompts for each image, following Samadh et al. (2023). The prompts are **reset** to the ImageNet trained prompts after each update.

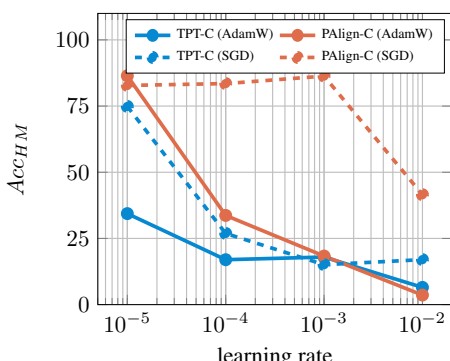

Figure 4: Performance of TPT-C and PAlign-C for CIFAR-10C/MNIST with AdamW and SGD optimizer on varying learning rates.

**TPT-C Shu et al. (2022)/ PAlign-C Samadh et al. (2023):** We create the continuous prompt update versions of TPT and PAlign as TPT-C and PAlign-C respectively. The only difference is that the prompts are continuously updated using the test stream of samples. If a sample is detected as reliable $C_d$ sample, the respective test time objectives are used to update the prompts. For this purpose, we vary the learning rate and optimizer to select the best optimizer for continuous prompt update. On performing experiments on CIFAR-10C/MNIST data, from Figure 4 we observe that SGD optimizer with learning rate $10^{-5}$ works the best for continuous prompt update and hence we use this for all the experiments of TPT-C and PAlign-C.

**(K+1)PC Li et al. (2023):** The vision encoder is updated using a (K+1) way prototypical cross entropy loss . The prototypes are updated using the test stream of samples. The learning rate is set to 0.001.

**TDA (Karmanov et al., 2024):** We use $\tau_t$ from the LDA based $C_d$ vs $C_u$ identifier to recognise the desired and undesired class samples. Following Karmanov et al. (2024), we set the shot capacity to 3 and the number of key-value caches is $C_d$ as we use the adapter only for desired class samples.

**DPE (Zhang et al., 2024):** We use the same LDA based $C_d$ vs $C_u$ identifier to recognise the desired and undesired class samples. We use the same hyperparameters presented in Zhang et al. (2024). A priority queue storing 3 visual features per class is used. The text and visual prototype residuals are updated with a learning rate of 0.0006 using AdamW optimizer.

**UniEnt:** We use the UniEnt objective in combination with LDA based class indentifier. The entropy minimization and maximization objectives are used for desired and undesired class samples respectively. The LayerNorm parameters are updated with a learning rate of 0.001 using SGD optimizer.

**ROSITA:** We use SGD optimizer with a learning rate of 0.001 to update the LayerNorm affine parameters of the Vision encoder. We set the size of score bank $\mathcal{S}$ to 512, number of neighbours $K$ to 5 and the size of $M_u$ is set to to 64.

## B.5 LIMITATIONS AND SCOPE FOR FUTURE WORK

Although ROSITA performs better than the baselines, in datasets where CIFAR-10C is and CIFAR-100C is, where it is hard to distinguish desired and undesired class samples, the FPR is still quite high, indicating that there is still significant scope for improvement. While in this work, we aim to identify the undesired class samples as "I don't know", in many practical applications these new classes can be of interest and need to be included in the desired classes. This incremental nature of TTA, where the set of desired classes keep growing, can be potentially explored in the future. Additional parameter choices such as adapters, LoRA can be explored for fine-tuning the model.

## C   ADDITIONAL ANALYSIS

In this section, in addition to the analysis done in Section 5, we study the robustness of the proposed method ROSITA more extensively, in the terms of (1) Error bars on different test data streams, (2) Role of the parameter $K$, the number of neighbours, (3) Analysis of the scores $s_t$ on using different combinations of the proposed loss components, (4) Effectiveness of LDA based Desired vs Undesired class identifier in comparison with simple thresholding, (5) Complexity Analysis of MaPLe backbone.

### C.1   ANALYSIS ON ERROR BARS

To study the robustness of our method for differently ordered test streams, we run ROSITA with five random seeds and report the Mean and Standard deviation of the $Acc_{HM}$ in Table 8 for CIFAR-10C/100C as $D_d$ and MNIST, SVHN, Tiny ImageNet, CIFAR-100C/10C as $D_u$ (corresponding to our results in Table 3 in the main paper). We observe that the variance in the performance of ROSITA is very low, reinforcing the robustness of the proposed method for different shuffled datasets and augmentations created.

Table 8: Performance (Mean and Standard deviation of $Acc_{HM}$) of ROSITA across 5 random seeds for CIFAR-10/100C as $D_d$ with 4 other datasets as $D_u$.

| $D_d \backslash D_u$ | MNIST | SVHN | Tiny | CIFAR-100/10C |
|---|---|---|---|---|
| CIFAR-10C | $84.07 \pm 0.023$ | $78.90 \pm 0.038$ | $80.10 \pm 0.014$ | $69.44 \pm 0.018$ |
| CIFAR-100C | $57.09 \pm 0.041$ | $47.90 \pm 0.047$ | $55.95 \pm 0.051$ | $48.10 \pm 0.024$ |

### C.2   ANALYSIS ON PARAMETER K

Table 9: Performance ($Acc_{HM}$) on varying $K$ with MNIST as $D_u$.

| $D_d$ | $|C_d|$ | $K$ | | | | | |
|---|---|---|---|---|---|---|---|
| | | 0 | 1 | 3 | 5 | 7 | 9 |
| CIFAR-10C | 10 | 80.97 | 83.9 | 84.32 | 84.17 | 84.10 | 84.02 |
| ImageNet-R | 200 | 64.32 | 83.65 | 83.87 | 83.53 | 83.39 | 83.42 |
| ImageNet-C | 1000 | 42.05 | 48.35 | 47.17 | 48.53 | 48.37 | 47.73 |

We vary the hyperparameter $K$ which represents the number of positives and negatives chosen in Equation 5 and 6 and report the results ($Acc_{HM}$) in Table 9. The size of the feature bank $\mathcal{M}_d$ is set as $N_d = K \times C_d$. $N_d$ increases with the number of classes as well as the number of neighbours $K$. We set $K$ to be 5 in all main results reported, which corresponds to feature bank size $N_d$ of 50, 1000, 5000 respectively for the datasets CIFAR-10C, ImageNet-R and ImageNet-C respectively. In Table 9, we abuse the notion $K = 0$ to correspond to the case where only the reliable pseudo label loss $\mathcal{L}_{Re}$ is used. The results show that even with $K = 1$, there is a significant improvement in $Acc_{HM}$ when compared to the case where $\mathcal{L}_D, \mathcal{L}_U$ is not used ($K = 0$). On further increasing $K$, we observe improvement only for the CIFAR-10C as $D_d$, but the performance is similar for ImageNet-R and ImageNet-C for higher values of $K$ as well. Further, we investigate this observation that the performance of ROSITA is similar on significantly varying $K$ or the feature bank size. For $K = 5$, we check the average number of positives actually selected for $L_D$ in Equation 5 for each of these datasets. We find this to be $4.1, 2.5$ and $1.5$ for CIFAR-10C, ImageNet-R and ImageNet-C respectively. This agrees with the results in Table 9 where $K$ of 3, 5 works better compared to 1 as more neighbours have common pseudo label, aiding the clustering of classes of interest. For CIFAR-10C and ImageNet-R, using $K < 5$ suffices and for ImageNet-C as only 1-2 neighbours are matched for majority of reliable desired class samples, setting $K = 1$ suffices. For practical purposes, this observation suggests that the buffer size for $M_d$ can indeed be reduced based on storage budget available depending on the application and device the model is deployed on. For e.g., if the memory budget available can store only upto 1000 features, $K$ can be set flexibly depending on the number of classes of interest. For ImageNet-C with 1000 classes, $K$ can be set to 1.

## C.3 Analysis of ReDUCe Loss components

We provide detailed results of Table 4 including all the five metrics in Table 10. Additionally, we visualise the histograms of the scores $s_t$ on using different combinations of the loss components of ReDUCe Loss in the Figures 5, 6, justifying their role in better discrimination of samples from $C_d$ and $C_u$.

Table 10: Detailed performance metrics analysing the ReDUCE Loss components.

| $\mathcal{L}_{Re}$ | $\mathcal{L}_D$ | $\mathcal{L}_U$ | CIFAR-10C/MNIST | | | | | ImageNet-R/MNIST | | | | |
|---|---|---|---|---|---|---|---|---|---|---|---|---|
| | | | AUC | FPR | $Acc_D$ | $Acc_U$ | $Acc_{HM}$ | AUC | FPR | $Acc_D$ | $Acc_U$ | $Acc_{HM}$ |
| ✗ | ✗ | ✗ | 91.91 | 85.04 | 60.82 | 99.77 | 75.57 | 91.27 | 91.09 | 55.67 | 99.90 | 71.50 |
| ✓ | ✗ | ✗ | 95.29 | 30.82 | 68.36 | 99.30 | 80.97 | 81.07 | 99.02 | 48.42 | 95.76 | 64.32 |
| ✗ | ✓ | ✗ | 95.23 | 28.91 | 66.93 | 98.52 | 79.71 | 87.73 | 94.67 | 51.13 | 98.34 | 67.28 |
| ✗ | ✗ | ✓ | 98.61 | 12.73 | 66.60 | 99.68 | 79.84 | 99.39 | 4.81 | 67.81 | 99.99 | 80.82 |
| ✗ | ✓ | ✓ | 99.27 | 4.15 | 67.76 | 99.73 | 80.69 | 99.48 | 4.40 | 69.38 | 99.98 | 81.92 |
| ✓ | ✓ | ✓ | 99.10 | 7.63 | 72.81 | 99.74 | 84.17 | 99.44 | 4.29 | 71.73 | 99.98 | 83.53 |

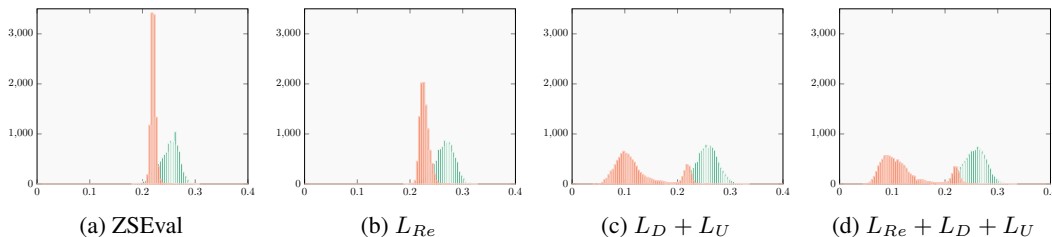

|     (a) ZSEval     |     (b) $L_{Re}$     |     (c) $L_D + L_U$     |     (d) $L_{Re} + L_D + L_U$     |

Figure 5: Histograms of $C_d$ and $C_u$ class scores for ZS-Eval and on using different loss components of the proposed ReDUCe loss on CIFAR-10C/MNIST dataset with CLIP.

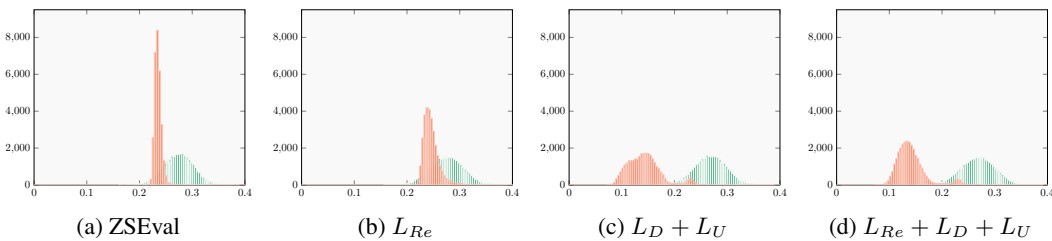

|     (a) ZSEval     |     (b) $L_{Re}$     |     (c) $L_D + L_U$     |     (d) $L_{Re} + L_D + L_U$     |

Figure 6: Histograms of $C_d$ and $C_u$ class scores for ZS-Eval and on using different loss components of the proposed ReDUCe loss on ImageNet-R/MNIST dataset with CLIP.

From Figure 5 and 6, we observe that, on using just $\mathcal{L}_{Re}$, the scores of $C_d$ and $C_u$ classes still sufficiently overlap, similar to the case of ZSEval. The performance purely depends on the quality of pseudo labels of the detected reliable desired class samples. In CIFAR-10C, as there are only 10 classes and given that ZSEval performance in CIFAR-10C is fairly good, it ensures good quality pseudo labels, hence resulting in overall better metrics on even using $\mathcal{L}_{Re}$ as shown in Table 10. ImageNet-R dataset inherently has more confusion as it is a 200-way classification problem. This naturally could result in lower quality pseudo labels, in turn degrading the performance compared to ZSEval. Alongside, using $\mathcal{L}_{Re}$ for desired class samples which are misclassified as undesired class samples increases the FPR and results in a decrease in metrics overall compared to ZSEval. On the other hand, using $\mathcal{L}_D$ and $\mathcal{L}_U$ separates the scores $s_t$ of samples from $C_d$ and $C_u$, resulting in two distinct peaks as seen in Figure 5 and 6, which in turn results in a significantly low FPR as reported in Table 10. Hence, the best results (Table 10) are obtained using the proposed ReDUCe loss where all the loss components aid each other to better discriminate the desired classes $C_d$ from $C_u$ (measured by AUC, FPR) and also improving the $C_d$-way accuracy ($Acc_D$) on desired classes.

## C.4 Comparison of different $C_d$ vs $C_u$ Class identifiers for Open-set TTA

To study the role of the $C_d$ vs $C_u$ class identifiers in Open-set Single Image TTA, we experiment with three class identifiers, on five datasets as $D_d$ with MNIST as $D_u$ using CLIP backbone.

**(1) Simple thresholding:** We set fixed thresholds $\tau_u, \tau_d$ to identify reliable samples from $C_d$ and $C_u$ classes respectively and $\tau_t$ to distinguish between $C_d$ and $C_u$ samples. We combine this class identifier with the ReDUCe loss of the proposed ROSITA framework.

**(2) Distribution Aware Filter (DAF) Gao et al. (2024) :** We adopt the Distribution Aware Filter proposed in UniEnt Gao et al. (2024), a very recent method on open-set TTA using CNNs, where they model the scores $s_t$ (similarity between image feature and source prototype) as a Gaussian Mixture Model for each batch. In our case, as we do single image TTA, we use a score bank as described in Section 2.4 as a proxy for the batch of samples, to estimate the parameters of the GMM. As it is a 2-component GMM, we identify a sample as a desired class sample if the probability $\pi(x_t)$ of the sample belonging to the desired classes(component with higher mean estimated) is greater than 0.5 or vice versa. The GMM based class identifier is defined as follows:

$$\hat{y} \begin{cases} \in C_d & \text{if } \pi(x_t) \geq 0.5 \\ \in C_u & \text{if } \pi(x_t) < 0.5 \end{cases} \tag{15}$$

We combine this class identifier with the Unified entropy objective and ReDUCe loss proposed by UniEnt Gao et al. (2024) and our proposed ROSITA framework respectively.

**(2) Linear Discriminant Analysis (LDA) based Li et al. (2023) :** As described in Section 2.4, we set $\tau_d$ to $\mu_d$ and $\tau_u$ to $\mu_u$ to identify reliable $C_d$ and $C_u$ samples to perform TTA. We set $\tau_t$ to $\mu_u$ to distinguish between $C_d$ and $C_u$ samples. The thresholds are estimated in an online manner using the score bank $\mathcal{S}$. The LDA based class identifier is defined as follows:

$$\hat{y} \begin{cases} \in C_d & \text{if } s_t \geq \tau_t^* \\ \in C_u & \text{if } s_t < \tau_t^* \end{cases} \tag{16}$$

We combine this class identifier with the Unified entropy objective and ReDUCe loss proposed by UniEnt Gao et al. (2024) and our proposed ROSITA framework respectively. The three thresholds for ReDUCe loss in Table 11 correspond to $\tau_u/\tau_t/\tau_d$ where $\tau_u$ and $\tau_d$ is used to identify reliable test samples and $\tau_t$ is used to distinguish between $C_d$ and $C_u$ samples. In the case of DAF with ReDUCe loss, we use the means $\mu_d^*$ and $\mu_*$ for the two gaussian mixture components to identify reliable samples.

Table 11: Comparison of $C_d$ vs $C_u$ class identifiers: MSP vs LDA vs (Distribution Aware Filter) DAF. The three thresholds for ReDUCe loss correspond to $\tau_u/\tau_t/\tau_d$ where $\tau_u$ and $\tau_d$ is used to identify reliable test samples and $\tau_t$ is used to distinguish between $C_d$ and $C_u$ samples. In the case of DAF with ReDUCe loss, we use the estimated means $\mu_d^*$ and $\mu_*$ of the two gaussian mixture components to identify reliable samples.

| $C_d$ vs $C_s$ | Threshold | Test-time objective | $D_u$: MNIST | | | | |
| | | | C-10C | C-100C | IN-C | IN-R | VisDA |
|---|---|---|---|---|---|---|---|
| MSP | 0.4/0.6/0.8 | ReDUCe | 43.44 | 34.42 | 1.20 | 77.12 | 88.49 |
| | 0.3/0.5/0.7 | | 33.70 | 32.60 | 1.74 | 80.29 | 50.87 |
| | 0.5/0.5/0.5 | | 22.82 | 37.41 | 1.91 | 30.90 | 32.31 |
| LDA | $s_t > \tau_t$ | UniEnt | 75.62 | 48.31 | 41.53 | 71.73 | 78.09 |
| DAF | $\pi(x_t) > 0.5$ | | 79.43 | 50.12 | 46.52 | 79.30 | 86.79 |
| LDA | $\mu_u/\tau_t/\mu_d$ | ReDUCe | **84.17** | **57.34** | **48.53** | **83.53** | 90.64 |
| DAF | $\mu_u^*/0.5/\mu_d^*$ | | 83.56 | 55.37 | 48.33 | 83.32 | **90.97** |

Our key observations based on the results in Table 11 are as follows:

**Fixed vs Dynamic Thresholds:** The performance of both, DAF and LDA based class identifier is significantly better than the simple thresholding case on adaptation using ReDUCe loss. The thresholds estimated in an online manner using the score bank $\mathcal{S}$ are more reliable than fixed

thresholds. The DAF and LDA based class identifier is able to better discriminate between $C_d$ and $C_u$ samples, resulting in better performance.

**UniEnt vs ReDUCe loss:** The performance on using ReDUCe loss (with either DAF or LDA class identifier) is significantly better than using the Unified entropy objective proposed in UniEnt Gao et al. (2024). The ReDUCe loss components aid each other to better discriminate the desired classes $C_d$ from $C_u$ (measured by AUC, FPR) and also improving the $C_d$-way accuracy ($Acc_D$) on desired classes.

**LDA vs DAF with ReDUCe loss:** The performance of LDA and DAF based class identifier perform very similarly when used in combination with ReDUCe loss. This suggests that ReDUCe loss in ROSITA is robust to the choice of a dynamically updating class identifier.

**Why is ReDUCe loss better than Unified entropy objective for Open-set TTA of VLMs?**

- Both LDA Li et al. (2023) and DAF Gao et al. (2024) were proposed for CNN based open-set TTA where a source model is trained on say clean data and is adapted to new domains, with the observation that the feature-prototype similarity scores $s_t$ can distinguish desired and undesired class samples. In the case of VLMs, the source model is trained on a large scale dataset and is adapted to potentially unseen/corrupted/covariate-shifted data. *The prior that the feature-prototype similarity scores $s_t$ can distinguish desired and undesired class samples does not translate to VLMs as the scores overlap significantly,* as observed in ZSEval histogram plots in Figures 3 5 6.

- In the case of CNNs, where the the initial scores are well separated and model has access to a batch of test samples at a time, UniEnt leverages this to further aid the separation of desired and undesired class samples in the batch through the UniEnt objective. In the case of VLMs, the scores are not well separated initially. This results in the means $\mu_d$ and $\mu_u$ in the case of LDA to be very close leading to misclassification of $C_d$ and $C_u$ class samples using the estimated threshold $\tau_t$. Similarly, in the case of DAF, the two components of GMM would not be very distinctive to well distinguish desired and undesired class samples. This misclassification can result in entropy minimization being applied on $C_u$ samples and entropy maximization on $C_d$ samples, which is undesirable. Employing UniEnt objective with several misclassified samples may not actually separate desired and undesired classes, as also empirically observed in Tables 1 2 3 (UniEnt has high FPR rate in general). Entropy maximization of $C_u$ samples does not explicitly enforce the separation of desired and undesired class samples in the feature space.

- The $L_D$ and $L_U$ loss components of ReDUCe loss explicitly enforce the separation desired and undesired class samples in the common VL latent space, while the $L_{Re}$ loss aims to only align the desired class samples to align with the text prototypes. With time, the model is adapted such that undesired class samples are away from the desired class samples and also the text prototypes. This ReDUCe loss addresses the challenges in single image open-set TTA in a holistic manner, resulting in better performance.

- On adopting UniEnt objective to single-image TTA, either entropy minimization or maximization loss would be active based on whether a test sample is identified as desired or undesired class sample, which is a limitation, as the objective cannot enforce distinction between the two types of features.

- In the case of CNNs, where the the initial scores are well separated and model has access to a batch of test samples at a time, UniEnt leverages this to further aid the separation of desired and undesired class samples in the batch through the UniEnt objective. In the case of VLMs, the scores are not well separated initially, hence the ReDUCe loss components (with the help of feature banks) is the driving force to better separate the desired and undesired class samples in the common latent space, resulting in lower FPR rates as a consequence.

## C.5  NEED FOR RELIABLE SAMPLES

To understand the role of selecting reliable samples for TTA, we do a simple experiment where we only use the threshold $\tau_t$ to distinguish between $C_d$ and $C_u$ samples. For all the samples with $s_t > \tau_t$ identified to belong to $C_d$, we perform TTA using $\mathcal{L}_{Re} + \mathcal{L}_D$ (Equation 5). Similarly, we use $L_U$ ( Equation 6) for all samples identified to belong to $C_u$ based on the criterion $s_t < \tau_t$. From the

Table 12: Performance of ROSITA using all samples vs only reliable samples for TTA.

| Thresholds | $D_u$: MNIST | | | | |
|---|---|---|---|---|---|
| $\tau_u/\tau_t/\tau_d$ | C-10C | C-100C | IN-C | IN-R | VisDA |
| $\tau_t/\tau_t/\tau_t$ | **84.99** | 55.16 | 44.05 | 83.28 | **91.24** |
| $\mu_u/\tau_t/\mu_d$ | 84.17 | **57.34** | **48.53** | **83.53** | 90.64 |

results in Table 12, we see that, for CIFAR-10C and VisDA, this case performs slightly better than our case(last row in Table 12) where TTA is performed only on reliable samples. CIFAR-10C and VisDA dataset have 10 and 12 classes of interest respectively. The zero shot performance of these datasets being good, as the class confusion is less, using all samples for TTA can be helpful. On the other hand, the classification in CIFAR-100C, ImageNet-C and ImageNet-R is harder, due the inherent confusion arising due to the large number of classes. Using non reliable test samples, with scores in the range $\mu_u < s_t < \mu_d$ can adversely affect the adaptation process. Hence, using only reliable samples for TTA performs better for these datasets as seen in Table 12). In a real world test time adaptation scenario, where we have no prior information about the difficulty of the classification task, in terms of severity of domain shift and class confusion, it is desirable to only use reliable samples for model updates.

### C.6    PERFORMANCE OF ROSITA WITH TIME

We plot the scores $s_t$ of samples from $C_d$ and $C_u$, and the best threshold $\tau_t$, with time in Figure 7a on using ROSITA. We observe that the scores $s_t$ for $C_d$ and $C_u$ samples become distinctive with time and the threshold estimated $\tau_t$ continuously tracks the changes in the scores $s_t$. Better discrimination of $C_d$ and $C_u$ samples aids the test time adaption process in ROSITA, resulting in a gradual improvement in the accuracy metrics as shown in Figure 7b. The metrics in Figure 7b are calculated based on the test samples seen until time $t$.

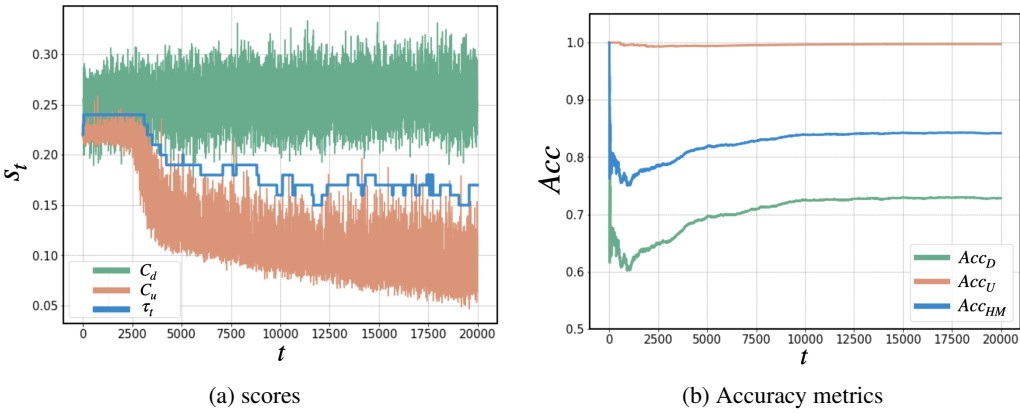

(a) scores                                                             (b) Accuracy metrics

Figure 7: Analysis of ROSITA on CIFAR-10C/MNIST: (a) Change in the scores of samples from $C_d$ and $C_u$ classes, the best threshold $\tau_t$ (based on LDA) with time $t$; (b) Accuracy metrics $Acc_D, Acc_U, Acc_{HM}$ measured for samples seen until time $t$. We see that the samples from $C_d$ and $C_u$ separate better with time. The accuracy metrics also improve with time.

**Unstable performance in the initial phase of TTA:** For the initial test samples ($t < 2500$), the scores of $C_d$ and $C_u$ samples overlap significantly(Figure 7a). The performance would be similar to the ZSEval(scores overlap at the beginning as the threshold identified $\tau_t$ classifies most $C_u$ samples accurately ($Acc_S$ is almost 100%), but misclassifies several desired class samples as $C_u$ (Figure 7b). A sample is predicted as one of the $C_d$ desired classes only if $s_t > \tau_t$. As several desired class samples are misclassified in this initial phase, this naturally leads to low $C_d$-way classification accuracy($Acc_D$) justifying the initial performance drop in Figure 7b. With time, as the model is updated with the proposed ReDUCe loss function, it better distinguishes $C_d$ and $C_u$ samples, separating their scores. For $t > 2500$, the model starts to accurately classify into $C_d$ or $C_u$, which in

turn results in gradual improvement of $Acc_D$ and $Acc_{HM}$ consequently. The instability (in the range $t < 1500$) can be attributed due to this initial learning process and also that the accuracy is measured on very less number of samples. In this case, as we are looking at single image TTA, the number of samples seen till time $t$ is also $t$ on which the accuracy metrics are measured and plotted, hence the oscillating nature, especially in the very early stages (say $t < 500$).

## C.7 EXTENSIVE PRELIMINARY ANALYSIS

Our initial experiments showed that updating LayerNorm parameters with simple entropy objective can effectively improve closed-set TTA performance. We illustrate this in Section 2.3 on CIFAR-10C dataset. Further, to justify our choice of updating LayerNorm parameters, we present the detailed experiments we conducted based on the following choices: (a) **Learnable parameters**: (1) Prompts, (2) Full network, (3) First Attention Block of ViT, (4) Last Attention Block of ViT (5) Prompts+LayerNorm(LN), (6) LayerNorm parameters (Zhao et al., 2023) (b) **Datasets**: In addition to CIFAR-10C (Section 2.3), we experiment with ImageNet-R, a relatively large scale dataset consisting of 30,000 images from 200 classes. (c) **Optimizer**: Along with SGD, we experiment with AdamW optimizer also used in [1], with varying learning rates on both CIFAR-10C and ImageNet-R dataset. We consistently observe that LayerNorm parameters is in general, a good choice to update the model.

Table 13: Accuracy on updating different parameter groups on CIFAR-10C and ImageNet-R datasets.

| Optimizer | Parameters | CIFAR-10C | | | | | ImageNet-R | | | | |
|---|---|---|---|---|---|---|---|---|---|---|---|
| | | $1e^{-6}$ | $1e^{-5}$ | $1e^{-4}$ | $1e^{-3}$ | $1e^{-2}$ | $1e^{-6}$ | $1e^{-5}$ | $1e^{-4}$ | $1e^{-3}$ | $1e^{-2}$ |
| SGD | Prompts | 73.40 | 31.04 | 12.53 | 11.18 | 10.19 | 73.97 | 74.17 | 74.71 | 25.68 | 10.63 |
| | Full | 10.48 | 10.44 | 9.99 | 10.00 | 10.01 | 14.18 | 7.19 | 0.65 | 0.65 | 0.42 |
| | First Block | 75.1 | 76.12 | 78.27 | 13.07 | 10.01 | 73.84 | 74.31 | 74.91 | 8.76 | 0.32 |
| | Last Block | 73.45 | 72.42 | 59.44 | 10.17 | 10.02 | 75.95 | 77.93 | 24.82 | 0.52 | 0.67 |
| | Prompts+LN | 73.82 | 46.77 | 24.71 | 10.24 | 10.18 | 73.76 | 75.09 | 76.35 | 28.72 | 11.74 |
| | LayerNorm | 74.35 | 76.61 | 80.41 | **84.58** | 11.69 | 74.13 | 74.35 | 75.23 | **76.92** | 33.07 |
| AdamW | Prompts | 72.40 | 18.6 | 12.83 | 10.04 | 10.08 | 74.4 | 75.17 | 27.93 | 6.82 | 4.37 |
| | Full | 10.32 | 10.03 | 10.00 | 10.00 | 9.97 | 14.83 | 0.95 | 0.28 | 0.52 | 0.66 |
| | First Block | 79.05 | 24.70 | 10.84 | 10.00 | 10.00 | 74.6 | 74.8 | 5.68 | 0.26 | 0.15 |
| | Last Block | 59.23 | 10.84 | 10.49 | 10.00 | 10.01 | 77.44 | 10.67 | 0.51 | 0.25 | 0.33 |
| | Prompts+LN | 75.01 | 72.10 | 21.92 | 13.33 | 10.01 | 74.52 | 76.45 | 12.99 | 8.87 | 5.55 |
| | LayerNorm | 76.10 | 81.57 | **85.9** | 85.27 | 10.03 | 73.96 | 75.64 | 78.28 | **78.81** | 31.47 |

## C.8 COMPLEXITY ANALYSIS

In Figure 8 and 9, we plot the GPU memory required and the time taken(secs/image) for TTA on each dataset using CLIP and MaPLe backbone respectively. The GPU memory and time taken scales with the number of classes for the prompt tuning baseline TPT. However, in ROSITA, the computational complexity is comparable to the ZS-Eval case. The text classifiers are obtained once and kept fixed throughout the adaptation process as in ZS-Eval. In ROSITA, we perform a forward pass of the image and its augmentation and one backward pass if a sample is categorized as reliable $C_d$ or $C_u$ sample.

For CLIP backbone, for ImageNet-C dataset with 1000 classes, ZSEval, TPT, TDA and ROSITA require 5.71 GB, 23.24 GB, 5.71 GB and 5.73 GB GPU memory to perform a single image based model update.

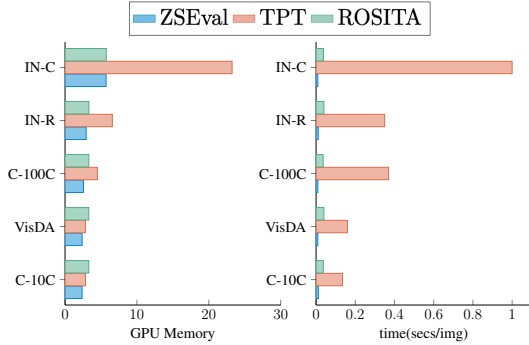

Figure 8: Complexity Analysis of different methods using CLIP backbone.

For MaPLe backbone, for ImageNet-C dataset with 1000 classes, ZSEval, PAlign and ROSITA require 5.94 GB, 29.12 GB and 5.98 GB GPU memory to perform a single image based model update. This makes the use of PAlign impractical and expensive for real time deployment in test scenarios, making it especially hard to port it on edge devices. The time taken to process a single image is 0.008s, 0.232s and 0.036s using ZSEval, PAlign and ROSITA respectively. Hence, **ROSITA** is computationally very efficient, similar to that of ZSEval.

This shows that **ROSITA** achieves the best trade off between memory and time complexity, being at par with ZSEval in terms of computational requirements while significantly outperforming ZSEval and the prompt tuning methods TPT and PAlign.

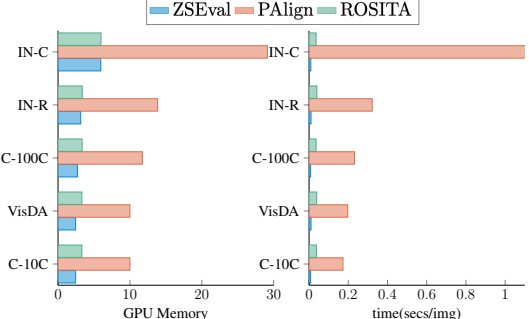

Figure 9: Complexity Analysis of different methods using MaPLe backbone.

**Memory buffer:** Prior prompt tuning methods like TPT Shu et al. (2022), Samadh et al. (2023) do not require any memory buffer. TDA Karmanov et al. (2024) requires a memory buffer of size $(|C_d| \times (3+2)) \times F$ to store 3 features per desired class in the positive cache and 2 features per class in the negative cache. DPE Zhang et al. (2024) requires a memory buffer of size $(|C_d| \times 3) \times F$ to store 3 features per desired class. ROSITA requires a memory buffer of size $(|C_d| \times 5 + 64) \times F$ to store 5 features per desired class and 64 features for the undesired classes. Here, F is the feature dimension. TDA, DPE and ROSITA are all memory efficient, requiring only about 10MB of additional memory even for a hard dataset like ImageNet-C with 1000 desired classes. However, *ROSITA makes best use of the buffer, storing desired sample features for retrieving positive neighbours and undesired sample features for retrieving negative neighbours, which in turn results in better performance compared to TDA and DPE as observed in Tables 1 2 3.*

## C.9   PERFORMANCE OF ROSITA ON LARGE VISION LANGUAGE BACKBONES

Here, in addition to CLIP ViT-B/16 Radford et al. (2021) and MAPLE Khattak et al. (2023) backbones, we perform experiments using large-scale Vision language backbones including CLIP ViT-L/14 by OpenAI Radford et al. (2021) and Open-CLIP ViT-L/14 Cherti et al. (2023) with CIFAR-10C/100C as $D_d$ and MNIST, SVHN, Tiny-ImageNet and CIFAR-100C/10C as $D_u$. From Table 14, we observe that ROSITA consistently outperforms even very recent baselines like TDA Karmanov et al. (2024), suggesting that the performance of ROSITA is agnostic to the choice of VL backbone.

Table 14: Comparison of ROSITA with prior methods on large scale Vision Language backbones.

| VL Backbone | Method | CIFAR-10C | | | | CIFAR-100C | | | |
|---|---|---|---|---|---|---|---|---|---|
| | | MNIST | SVHN | Tiny | C-100C | MNIST | SVHN | Tiny | C-10C |
| CLIP ViT-L/14 | ZSEval | 83.94 | 74.54 | 80.16 | 72.32 | 56.29 | 52.35 | 53.25 | 49.89 |
| | (K+1)PC | 85.43 | 80.60 | 81.65 | 71.90 | 64.14 | 55.18 | 54.53 | 47.90 |
| | TDA | 84.91 | 76.87 | 81.07 | 74.23 | 59.11 | 55.25 | 55.44 | 52.48 |
| | ROSITA | **89.46** | **83.42** | **83.61** | **75.63** | **65.41** | **60.31** | **57.55** | **54.66** |
| Open-CLIP ViT-L/14 | ZSEval | 80.64 | 76.90 | 84.10 | 75.40 | 62.96 | 59.38 | 61.10 | 59.57 |
| | (K+1)PC | 85.84 | 82.42 | 84.99 | 75.70 | 70.14 | 63.36 | 60.56 | 59.43 |
| | TDA | 80.57 | 77.92 | 84.60 | 75.79 | 64.90 | 60.70 | 62.01 | 61.20 |
| | ROSITA | **89.04** | **82.98** | **85.55** | **76.62** | **70.54** | **63.84** | **62.57** | **61.84** |

## D    ADDITIONAL EXPERIMENTS

In addition to the results presented in the main paper, we perform additional experiments supporting the claims made and for more comprehensive understanding of the analysis presented in Section 5.

### D.1    EXPERIMENTS USING DIFFERENT CORRUPTION TYPES

To evaluate the robustness of our method across different domains, we do additional experiments with *impulse noise* , *motion blur* and *jpeg compression* corruptions from the corruption categories per-pixel noise, blurring and digital transforms respectively and report the results here. From Table 15, Table 16 and Table 17, we observe that ROSITA either outperforms or at par with prior methods in most cases even on using the same set of hyperparameters. This demonstrates its robustness across a variety of corruption types.

Table 15: Results on CIFAR-10C/100C (Impulse Noise) as $D_d$ with other $D_u$.

| | | Method | MNIST | | | SVHN | | | Tiny-ImageNet | | | CIFAR-100C/10-C | | |
|---|---|---|---|---|---|---|---|---|---|---|---|---|---|---|
| | | | AUC ↑ | FPR ↓ | HM ↑ | AUC ↑ | FPR ↓ | HM ↑ | AUC ↑ | FPR ↓ | HM ↑ | AUC ↑ | FPR ↓ | HM ↑ |
| C-10C (Impulse noise) | CLIP | ZS-Eval | 86.34 | 97.77 | 57.67 | 84.40 | 79.43 | 56.80 | 88.97 | 31.86 | 61.11 | 78.61 | 67.88 | 54.40 |
| | | TPT | 86.35 | 97.83 | 59.80 | 84.43 | 79.52 | 58.97 | 88.96 | 31.99 | 64.48 | 78.60 | 68.24 | 56.38 |
| | | TPT-C | 62.34 | 87.66 | 39.90 | 59.71 | 83.29 | 35.42 | 81.30 | 38.59 | 37.02 | 66.22 | 89.92 | 30.86 |
| | | ROSITA | 98.87 | 9.43 | 71.31 | 82.85 | 56.82 | 61.03 | 93.36 | 21.47 | 64.47 | 78.69 | 69.45 | 57.87 |
| | MAPLE | ZS-Eval | 91.10 | 76.09 | 64.01 | 92.98 | 45.28 | 63.66 | 83.77 | 44.44 | 60.93 | 79.22 | 65.26 | 57.49 |
| | | PAlign | 91.10 | 76.01 | 65.76 | 93.00 | 45.13 | 65.28 | 83.78 | 44.42 | 62.75 | 79.22 | 65.24 | 58.80 |
| | | PAlign-C | 92.43 | 63.39 | 63.61 | 92.92 | 45.86 | 64.50 | 83.36 | 45.74 | 60.83 | 79.30 | 64.47 | 57.00 |
| | | ROSITA | 98.80 | 6.10 | 71.79 | 95.39 | 28.06 | 72.13 | 84.92 | 45.35 | 65.30 | 80.49 | 65.57 | 61.63 |
| C-100C (Impulse noise) | CLIP | ZS-Eval | 70.48 | 99.17 | 25.08 | 51.12 | 96.44 | 25.69 | 59.90 | 67.18 | 27.72 | 53.51 | 94.97 | 25.16 |
| | | TPT | 70.56 | 99.17 | 25.26 | 51.21 | 96.38 | 26.26 | 59.91 | 67.09 | 28.36 | 53.53 | 94.94 | 25.63 |
| | | TPT-C | 57.65 | 93.07 | 8.71 | 79.28 | 57.07 | 2.74 | 90.40 | 22.60 | 5.71 | 50.26 | 95.34 | 3.26 |
| | | ROSITA | 36.47 | 99.96 | 20.98 | 24.17 | 99.77 | 18.99 | 53.57 | 79.85 | 26.27 | 58.02 | 94.15 | 29.75 |
| | MAPLE | ZS-Eval | 69.29 | 89.49 | 33.66 | 81.03 | 73.94 | 34.99 | 49.57 | 84.71 | 26.09 | 57.84 | 94.44 | 29.34 |
| | | PAlign | 69.31 | 89.54 | 33.74 | 81.05 | 73.98 | 34.96 | 49.60 | 84.63 | 25.81 | 57.84 | 94.48 | 29.53 |
| | | PAlign-C | 71.14 | 73.63 | 34.38 | 82.08 | 68.24 | 35.11 | 47.27 | 87.87 | 25.95 | 57.79 | 93.54 | 30.73 |
| | | ROSITA | 95.38 | 8.80 | 43.06 | 80.25 | 41.21 | 34.88 | 42.77 | 97.15 | 19.70 | 49.73 | 96.72 | 12.62 |

Table 16: Results on CIFAR-10C/100C(Motion blur) as $D_d$ with other $D_u$.

| | | Method | MNIST | | | SVHN | | | Tiny-ImageNet | | | CIFAR-100C/10-C | | |
|---|---|---|---|---|---|---|---|---|---|---|---|---|---|---|
| | | | AUC ↑ | FPR ↓ | HM ↑ | AUC ↑ | FPR ↓ | HM ↑ | AUC ↑ | FPR ↓ | HM ↑ | AUC ↑ | FPR ↓ | HM ↑ |
| C-10C (Motion blur) | CLIP | ZS-Eval | 97.73 | 2.75 | 73.69 | 96.40 | 18.34 | 73.82 | 95.25 | 15.75 | 74.27 | 79.57 | 70.08 | 62.86 |
| | | TPT | 97.72 | 2.68 | 74.15 | 96.39 | 18.16 | 74.42 | 95.23 | 15.72 | 75.03 | 79.56 | 69.86 | 63.25 |
| | | TPT-C | 80.73 | 86.28 | 63.74 | 62.09 | 62.52 | 42.19 | 80.76 | 51.66 | 48.04 | 55.66 | 97.04 | 37.53 |
| | | ROSITA | 99.90 | 0.04 | 81.87 | 96.50 | 21.55 | 77.47 | 96.58 | 13.65 | 77.44 | 82.03 | 65.95 | 66.96 |
| | MAPLE | ZS-Eval | 96.52 | 18.33 | 78.68 | 97.08 | 14.78 | 78.15 | 88.45 | 33.15 | 71.19 | 84.00 | 57.94 | 66.93 |
| | | PAlign | 96.51 | 18.37 | 78.92 | 97.08 | 14.82 | 78.38 | 88.45 | 33.13 | 71.73 | 83.99 | 57.99 | 67.15 |
| | | PAlign-C | 97.17 | 13.47 | 78.49 | 96.89 | 15.87 | 78.09 | 88.80 | 32.94 | 72.09 | 84.29 | 56.80 | 67.40 |
| | | ROSITA | 98.49 | 10.01 | 83.26 | 92.61 | 44.87 | 78.93 | 87.48 | 38.23 | 73.24 | 84.27 | 57.60 | 70.67 |
| C-100C (Motion blur) | CLIP | ZS-Eval | 93.08 | 58.92 | 48.17 | 83.63 | 81.33 | 46.04 | 79.34 | 53.56 | 48.53 | 64.03 | 91.54 | 41.63 |
| | | TPT | 93.06 | 59.87 | 48.18 | 83.61 | 81.56 | 45.54 | 79.29 | 53.76 | 48.26 | 64.02 | 91.63 | 41.25 |
| | | TPT-C | 66.77 | 98.77 | 19.96 | 29.69 | 99.94 | 11.39 | 69.25 | 62.87 | 17.10 | 53.22 | 94.57 | 13.59 |
| | | ROSITA | 98.93 | 6.79 | 55.49 | 89.39 | 37.86 | 48.50 | 90.20 | 31.61 | 55.05 | 65.30 | 91.59 | 42.54 |
| | MAPLE | ZS-Eval | 81.21 | 80.28 | 45.66 | 89.04 | 60.73 | 46.98 | 60.84 | 80.63 | 40.60 | 64.01 | 90.18 | 42.30 |
| | | PAlign | 81.20 | 80.52 | 44.52 | 89.03 | 61.01 | 45.76 | 60.84 | 80.64 | 40.03 | 64.01 | 90.26 | 41.26 |
| | | PAlign-C | 82.72 | 68.08 | 49.92 | 90.48 | 53.83 | 51.87 | 62.00 | 82.85 | 41.66 | 64.47 | 89.05 | 43.58 |
| | | ROSITA | 97.12 | 7.78 | 57.30 | 85.13 | 56.16 | 49.89 | 63.85 | 80.20 | 42.65 | 62.55 | 94.62 | 41.54 |

### D.2    EXPERIMENTS USING CLIP VIT-B32 AND CLIP RESNET50 ARCHITECTURES

To test the performance of ROSITA and prior methods across different architectures, we perform additional experiments using CLIP ViT-B/32 and CLIP ResNet50 models. In CLIP ResNet50 model,

Table 17: Results on CIFAR-10C/100C(JPEG Compression) as $D_d$ with other $D_u$.

| | | Method | MNIST | | | SVHN | | | Tiny-ImageNet | | | CIFAR-100C/10-C | | |
|---|---|---|---|---|---|---|---|---|---|---|---|---|---|---|
| | | | AUC ↑ | FPR ↓ | HM ↑ | AUC ↑ | FPR ↓ | HM ↑ | AUC ↑ | FPR ↓ | HM ↑ | AUC ↑ | FPR ↓ | HM ↑ |
| C-10C (JPEG) | CLIP | ZS-Eval | 68.16 | 100.00 | 53.92 | 67.04 | 99.93 | 55.69 | 79.44 | 65.02 | 59.66 | 73.65 | 85.60 | 56.30 |
| | | TPT | 68.07 | 100.00 | 54.16 | 66.97 | 99.93 | 56.06 | 79.37 | 65.11 | 60.09 | 73.64 | 85.58 | 56.87 |
| | | TPT-C | 68.28 | 99.37 | 53.12 | 54.76 | 98.97 | 35.64 | 66.70 | 72.20 | 39.02 | 59.82 | 94.78 | 32.78 |
| | | ROSITA | 81.83 | 58.81 | 60.34 | 82.85 | 61.38 | 61.87 | 95.06 | 15.84 | 67.87 | 71.19 | 86.62 | 51.98 |
| | MAPLE | ZS-Eval | 95.15 | 33.39 | 69.72 | 95.96 | 22.02 | 69.73 | 86.64 | 36.79 | 65.68 | 79.26 | 68.19 | 60.10 |
| | | PAlign | 95.13 | 33.57 | 69.62 | 95.95 | 22.01 | 69.31 | 86.63 | 36.82 | 65.62 | 79.26 | 68.18 | 59.86 |
| | | PAlign-C | 96.53 | 20.14 | 70.50 | 95.94 | 21.51 | 70.01 | 87.38 | 35.07 | 66.42 | 79.85 | 66.17 | 61.11 |
| | | ROSITA | 99.28 | 5.71 | 76.74 | 95.54 | 29.06 | 72.86 | 89.88 | 31.12 | 68.78 | 80.69 | 61.64 | 62.23 |
| CIFAR-100C (JPEG) | CLIP | ZS-Eval | 50.88 | 100.00 | 32.27 | 39.25 | 100.00 | 26.41 | 48.65 | 95.60 | 29.92 | 53.51 | 95.59 | 32.48 |
| | | TPT | 50.78 | 100.00 | 32.38 | 39.18 | 100.00 | 26.48 | 48.55 | 95.60 | 29.86 | 53.49 | 95.57 | 32.70 |
| | | TPT-C | 12.11 | 100.00 | 3.32 | 10.05 | 99.98 | 2.45 | 63.07 | 90.01 | 9.49 | 52.23 | 95.05 | 6.33 |
| | | ROSITA | 29.10 | 100.00 | 22.83 | 35.58 | 99.94 | 23.50 | 50.76 | 94.76 | 31.64 | 53.96 | 96.18 | 30.39 |
| | MAPLE | ZS-Eval | 78.86 | 80.60 | 37.60 | 87.72 | 61.14 | 39.18 | 58.31 | 80.75 | 34.03 | 54.50 | 95.49 | 34.02 |
| | | PAlign | 78.82 | 80.92 | 36.62 | 87.69 | 61.37 | 38.01 | 58.29 | 80.79 | 33.17 | 54.49 | 95.52 | 32.96 |
| | | PAlign-C | 81.85 | 63.37 | 40.87 | 89.96 | 49.09 | 41.89 | 59.33 | 81.48 | 33.84 | 53.82 | 95.17 | 33.28 |
| | | ROSITA | 97.68 | 7.87 | 46.51 | 92.14 | 34.44 | 42.71 | 66.63 | 75.00 | 37.43 | 51.33 | 96.68 | 25.41 |

we finetune the BatchNorm parameters instead of LayerNorm. We observe that the performance improvement of ROSITA with respect to the baselines is agnostic to the model architecture of the VLM.

Table 18: Results on CIFAR-10C/100C as $D_d$ with four other $D_u$ datasets.

| | | Method | MNIST | | | SVHN | | | Tiny-ImageNet | | | CIFAR-100C/10-C | | |
|---|---|---|---|---|---|---|---|---|---|---|---|---|---|---|
| | | | AUC ↑ | FPR ↓ | HM ↑ | AUC ↑ | FPR ↓ | HM ↑ | AUC ↑ | FPR ↓ | HM ↑ | AUC ↑ | FPR ↓ | HM ↑ |
| CIFAR-10C | ViT-B/32 | ZS-Eval | 96.58 | 18.94 | 73.20 | 92.01 | 43.95 | 71.35 | 91.55 | 24.72 | 72.19 | 79.27 | 69.32 | 64.06 |
| | | TPT | 96.55 | 19.44 | 73.96 | 91.97 | 44.31 | 71.96 | 91.54 | 24.81 | 73.61 | 79.25 | 69.48 | 64.59 |
| | | TPT-C | 63.79 | 99.97 | 50.48 | 55.96 | 99.30 | 40.63 | 78.71 | 52.30 | 43.31 | 57.83 | 93.11 | 42.47 |
| | | ROSITA | 99.14 | 3.84 | 81.65 | 93.78 | 33.45 | 75.18 | 98.86 | 4.14 | 80.91 | 80.28 | 64.17 | 64.34 |
| | RN50 | ZS-Eval | 36.73 | 100.00 | 31.49 | 59.79 | 99.07 | 41.01 | 84.64 | 36.21 | 54.61 | 67.63 | 87.30 | 45.19 |
| | | TPT | 37.26 | 100.00 | 32.18 | 60.25 | 99.03 | 41.95 | 84.76 | 36.07 | 56.41 | 67.62 | 87.37 | 45.98 |
| | | TPT-C | 14.06 | 98.57 | 5.46 | 36.98 | 93.76 | 19.11 | 73.60 | 62.60 | 22.87 | 51.23 | 91.64 | 19.69 |
| | | ROSITA | 62.45 | 99.87 | 47.63 | 96.30 | 23.90 | 65.52 | 96.51 | 11.03 | 59.34 | 68.30 | 83.64 | 49.11 |
| CIFAR-100C | ViT-B/32 | ZS-Eval | 89.17 | 61.01 | 46.11 | 78.17 | 79.92 | 44.59 | 72.58 | 61.21 | 45.65 | 64.29 | 90.53 | 41.44 |
| | | TPT | 89.08 | 61.15 | 45.99 | 78.06 | 80.11 | 44.78 | 72.57 | 61.24 | 46.25 | 64.31 | 90.47 | 41.65 |
| | | TPT-C | 61.66 | 99.96 | 17.97 | 30.50 | 89.96 | 11.55 | 83.18 | 82.01 | 11.79 | 53.52 | 92.74 | 9.34 |
| | | ROSITA | 94.34 | 23.99 | 57.14 | 90.26 | 45.33 | 51.60 | 91.22 | 30.17 | 56.02 | 68.33 | 86.03 | 44.57 |
| | RN50 | ZS-Eval | 23.47 | 100.00 | 14.27 | 37.73 | 99.91 | 20.84 | 65.59 | 61.52 | 27.77 | 54.28 | 94.77 | 22.18 |
| | | TPT | 23.88 | 100.00 | 14.17 | 38.18 | 99.91 | 20.49 | 65.80 | 61.22 | 27.39 | 54.30 | 94.82 | 21.81 |
| | | TPT-C | 24.35 | 97.90 | 2.32 | 13.57 | 99.96 | 2.44 | 83.84 | 44.88 | 4.17 | 53.54 | 95.29 | 3.84 |
| | | ROSITA | 23.73 | 100.00 | 15.27 | 66.59 | 73.78 | 28.34 | 73.04 | 60.32 | 26.57 | 54.30 | 93.50 | 23.52 |

Table 19: Results with ImageNet-R/C as $D_d$ with MNIST and SVHN as $D_u$.

| | Method | IN-C/MNIST | | | IN-C/SVHN | | | IN-R/MNIST | | | IN-R/SVHN | | |
|---|---|---|---|---|---|---|---|---|---|---|---|---|---|
| | | AUC ↑ | FPR ↓ | HM ↑ | AUC ↑ | FPR ↓ | HM ↑ | AUC ↑ | FPR ↓ | HM ↑ | AUC ↑ | FPR ↓ | HM ↑ |
| ViT-B/32 | ZS-Eval | 89.17 | 61.01 | 46.11 | 78.17 | 79.92 | 44.59 | 72.58 | 61.21 | 45.65 | 64.29 | 90.53 | 41.44 |
| | TPT | 89.08 | 61.15 | 45.99 | 78.06 | 80.11 | 44.78 | 72.57 | 61.24 | 46.25 | 64.31 | 90.47 | 41.65 |
| | TPT-C | 61.66 | 99.96 | 17.97 | 30.50 | 89.96 | 11.55 | 83.18 | 82.01 | 11.79 | 53.52 | 92.74 | 9.34 |
| | ROSITA | 94.34 | 23.99 | 57.14 | 90.26 | 45.33 | 51.60 | 91.22 | 30.17 | 56.02 | 68.33 | 86.03 | 44.57 |
| RN50 | ZS-Eval | 91.15 | 61.44 | 17.56 | 92.37 | 43.01 | 19.23 | 87.39 | 98.23 | 57.87 | 92.34 | 55.18 | 60.40 |
| | TPT | 91.69 | 58.09 | 18.18 | 92.74 | 40.72 | 20.21 | 87.50 | 98.16 | 58.68 | 92.39 | 54.97 | 61.41 |
| | TPT-C | 95.00 | 10.45 | 1.74 | 29.09 | 99.98 | 1.31 | 71.95 | 97.61 | 37.79 | 75.25 | 78.47 | 41.85 |
| | ROSITA | 99.60 | 1.26 | 22.58 | 98.91 | 4.96 | 23.03 | 99.55 | 2.77 | 69.46 | 99.67 | 1.81 | 70.53 |

Table 20: Results with VisDA as $D_d$ with MNIST and SVHN as $D_u$ datasets.

| | Method | VisDA/MNIST | | | VisDA/SVHN | | |
|---|---|---|---|---|---|---|---|
| | | AUC ↑ | FPR ↓ | HM ↑ | AUC ↑ | FPR ↓ | HM ↑ |
| ViT-B/32 | ZS-Eval | 89.10 | 95.57 | 73.85 | 85.54 | 80.62 | 71.93 |
| | TPT | 89.06 | 95.61 | 74.05 | 85.49 | 80.72 | 72.11 |
| | TPT-C | 66.98 | 99.75 | 62.89 | 17.01 | 99.83 | 13.62 |
| | ROSITA | 99.17 | 4.50 | 87.83 | 97.35 | 16.56 | 84.89 |
| RN50 | ZS-Eval | 67.19 | 100.00 | 61.47 | 81.59 | 97.46 | 68.41 |
| | TPT | 67.28 | 100.00 | 61.60 | 81.60 | 97.43 | 68.62 |
| | TPT-C | 6.24 | 100.00 | 5.55 | 10.72 | 100.00 | 15.79 |
| | ROSITA | 78.57 | 99.96 | 66.89 | 98.44 | 8.06 | 79.87 |

## D.3 OPEN SET SINGLE IMAGE CTTA EXPERIMENTS

Here, we report the detailed corruption-wise results presented in Table 5. In addition, we evaluate the performance of ROSITA in comparison with prior methods more extensively here. We present the 15 corruptions of CIFAR-10C sequentially as $D_d$, one sample at a time along with different datasets for $C_u$ samples, namely MNIST, SVHN, Tiny ImageNet, CIFAR-100C and report the results in Table 21. We observe that the improvement in performance of ROSITA is agnostic to model architecture, challenging scenarios including different combinations of $D_d$ (continuously changing domains) and $D_u$ datasets.

Table 21: Results on Openworld Single Image Continuous Test Time Adaptation(CTTA) for CIFAR-10C (15 corruptions shown sequentially) as $D_d$ with other $D_u$ datasets.

| Du/Dd | Model | Method | gaussian | shot | impulse | defocus | glass | motion | zoom | snow | frost | fog | brightness | contrast | elastic | pixelate | jpeg | Mean |
|---|---|---|---|---|---|---|---|---|---|---|---|---|---|---|---|---|---|---|
| CIFAR-10C/MNIST | CLIP | ZS-Eval | 43.21 | 47.74 | 57.68 | 75.43 | 38.56 | 73.91 | 76.94 | 75.56 | 79.38 | 74.36 | 84.88 | 67.36 | 55.61 | 60.56 | 53.82 | 64.33 |
| | | TPT | 43.15 | 47.66 | **57.70** | 75.36 | 38.22 | 73.70 | 76.84 | 75.49 | 79.32 | 74.80 | 84.82 | 67.46 | 55.50 | 60.40 | 53.48 | 64.26 |
| | | TPT-C | 30.06 | 25.92 | 31.05 | 52.71 | 20.88 | 53.08 | 21.61 | 26.83 | 38.80 | 38.88 | 37.40 | 33.83 | 35.26 | 3.53 | | 33.05 |
| | | ROSITA | **43.35** | **48.21** | 57.04 | **78.01** | **43.29** | **77.48** | **80.16** | **76.84** | **80.15** | **76.26** | **86.33** | **73.44** | **60.35** | **61.55** | **60.38** | **66.86** |
| | MAPLE | ZS-Eval | 42.33 | 44.71 | 64.00 | 78.78 | 45.90 | 78.69 | 81.12 | 82.56 | 84.79 | 78.13 | 88.87 | 67.94 | 63.87 | 51.63 | 69.77 | 68.21 |
| | | PAlign | 42.95 | 44.22 | 64.85 | 77.36 | 44.70 | 78.44 | 80.16 | 82.46 | 83.47 | 77.25 | 88.29 | 65.49 | 64.34 | 51.73 | 67.53 | 67.55 |
| | | PAlign-C | 42.97 | 45.32 | 63.98 | 78.79 | 48.07 | 78.42 | 81.09 | 83.88 | 85.21 | 77.38 | 89.09 | 69.90 | 66.22 | 56.59 | 70.01 | 69.13 |
| | | ROSITA | **43.51** | **49.92** | **64.87** | **78.98** | **54.56** | **80.58** | **84.04** | **87.27** | **89.09** | **84.11** | **93.02** | **78.60** | **74.02** | **71.64** | **75.30** | **73.97** |
| CIFAR-10C/SVHN | CLIP | ZS-Eval | 42.86 | 47.15 | 56.79 | 75.11 | 41.57 | 74.03 | 76.65 | 74.07 | 77.73 | 73.66 | 83.01 | 68.03 | 54.80 | 59.66 | 55.58 | 64.05 |
| | | TPT | 42.82 | 47.10 | 56.82 | 74.98 | 41.49 | 73.88 | 76.64 | 74.05 | 77.67 | 73.93 | 82.95 | 68.32 | 54.70 | 59.60 | 55.51 | 64.03 |
| | | TPT-C | 37.26 | 34.53 | 39.45 | 62.23 | 30.72 | 55.30 | 62.65 | 45.74 | 47.70 | 50.35 | 55.42 | 57.01 | 43.26 | 45.32 | 29.64 | 46.44 |
| | | ROSITA | **43.08** | **47.99** | **57.62** | **76.73** | **42.35** | **74.99** | **78.59** | **76.34** | **78.54** | 72.00 | **83.58** | **68.93** | **60.21** | **60.08** | **57.86** | **65.26** |
| | MAPLE | ZS-Eval | 45.34 | 50.19 | 63.65 | 78.24 | 52.00 | 78.13 | 80.62 | 83.57 | 85.00 | 77.77 | 88.80 | 67.55 | 63.51 | 55.23 | 69.73 | 69.29 |
| | | PAlign | **45.74** | 50.29 | 64.35 | 76.99 | 51.50 | 77.97 | 79.89 | 83.16 | 83.63 | 76.89 | 88.47 | 65.56 | 64.10 | 55.91 | 67.70 | 68.81 |
| | | PAlign-C | 45.36 | 50.36 | 63.83 | 78.19 | 51.55 | 77.84 | 80.50 | 83.05 | 84.42 | 76.82 | 88.15 | 71.57 | 65.50 | 55.01 | 70.04 | 69.48 |
| | | ROSITA | 45.51 | **50.99** | **64.73** | **78.36** | **53.10** | **78.74** | **80.87** | **83.79** | **85.18** | **78.47** | **88.71** | 70.78 | **66.70** | **59.28** | **71.18** | **70.43** |
| CIFAR-10C/Tiny | CLIP | ZS-Eval | 49.41 | 52.96 | 61.09 | 76.40 | 49.23 | 74.28 | 77.36 | 74.49 | 77.39 | 73.92 | 81.34 | 70.26 | 60.29 | 59.40 | 59.67 | 66.50 |
| | | TPT | 49.43 | 52.97 | 61.07 | 76.41 | 49.13 | 74.27 | 77.36 | 74.63 | 77.43 | 74.05 | 81.49 | 70.14 | 60.16 | 59.28 | 59.66 | 66.50 |
| | | TPT-C | 49.64 | 51.56 | 59.10 | 74.35 | 47.37 | 66.65 | 71.56 | 60.46 | 62.19 | 63.91 | 69.60 | 63.85 | 55.65 | 52.31 | 42.58 | 59.38 |
| | | ROSITA | **49.64** | **53.56** | **61.64** | **77.02** | **50.23** | **76.09** | **79.22** | **78.05** | **79.34** | **76.84** | **84.55** | **73.65** | **65.87** | 58.86 | **68.76** | **68.89** |
| | MAPLE | ZS-Eval | 44.18 | 47.30 | 60.94 | 71.71 | 49.99 | 71.18 | 73.40 | 76.15 | 76.76 | 71.56 | 80.22 | 64.44 | 61.51 | 55.67 | 65.69 | 64.71 |
| | | PAlign | 44.17 | 46.35 | 61.56 | 70.27 | 48.90 | 70.63 | 72.46 | 75.57 | 75.32 | 70.66 | 79.65 | 62.53 | 62.15 | 56.28 | 63.13 | 63.98 |
| | | PAlign-C | **44.38** | **48.00** | 61.09 | 72.15 | 49.94 | 72.06 | 74.47 | 76.10 | **77.67** | 72.13 | 80.51 | 66.68 | 61.75 | 55.69 | 66.51 | 65.28 |
| | | ROSITA | 44.29 | 47.93 | **61.59** | **72.35** | **51.11** | **72.20** | **74.47** | **76.34** | 77.45 | **72.89** | **80.82** | **66.70** | **62.81** | **57.72** | **67.00** | **65.71** |
| CIFAR-10C/CIFAR-100C | CLIP | ZS-Eval | 40.48 | 44.50 | 54.34 | 67.17 | 40.46 | 62.85 | 68.16 | 68.90 | 70.68 | 65.22 | 76.26 | 62.16 | 51.48 | 48.42 | 56.23 | 58.49 |
| | | TPT | 40.43 | 44.45 | 54.32 | 67.13 | 40.40 | 62.89 | 68.14 | 68.90 | 70.71 | 65.17 | 76.24 | 62.13 | 51.41 | 48.46 | 56.31 | 58.47 |
| | | TPT-C | 27.80 | 26.46 | 33.01 | 40.72 | 28.05 | 38.78 | 42.05 | 41.90 | 43.91 | 39.15 | 45.80 | 41.50 | 37.11 | 32.71 | 39.69 | 37.24 |
| | | ROSITA | **40.66** | **45.15** | **55.01** | **67.31** | **41.07** | **63.12** | **68.54** | **69.58** | **71.09** | **66.23** | **76.34** | **63.89** | **54.15** | 48.23 | **57.08** | **59.16** |
| | MAPLE | ZS-Eval | 41.99 | 45.82 | 57.50 | 69.19 | 44.03 | 66.86 | 70.43 | **71.81** | 73.33 | 68.32 | 76.95 | 64.18 | 56.74 | 49.81 | 60.15 | 61.14 |
| | | PAlign | 41.93 | 45.16 | 57.81 | 68.04 | 42.44 | 66.54 | 69.56 | 71.35 | 71.78 | 67.46 | 76.70 | 62.17 | 56.98 | 49.86 | 58.22 | 60.40 |
| | | PAlign-C | 41.86 | 45.80 | 57.51 | 69.78 | 46.17 | **67.73** | **71.47** | 71.03 | 74.00 | 68.98 | 77.61 | 65.53 | 57.08 | 52.17 | 61.17 | 61.86 |
| | | ROSITA | **42.13** | **46.09** | **58.00** | 69.48 | 45.33 | 67.44 | 71.00 | 71.00 | 73.31 | **69.42** | **78.37** | 65.55 | 57.32 | 53.52 | 60.85 | 61.92 |

# E DETAILED EXPERIMENTAL RESULTS

Here, we report in detail all the metrics (Section 3), namely AUC, FPR, $Acc_W$, $Acc_S$, $Acc_{HM}$ of the main results presented in Table 1, Table 3.

Table 22: Detailed results using CIFAR-10C as $D_d$ with MNIST and SVHN as $D_u$.

| | Method | CIFAR-10C/MNIST | | | | | CIFAR-10C/SVHN | | | | |
|---|---|---|---|---|---|---|---|---|---|---|---|
| | | AUC | FPR | $Acc_D$ | $Acc_U$ | $Acc_{HM}$ | AUC | FPR | $Acc_D$ | $Acc_U$ | $Acc_{HM}$ |
| CLIP | ZS-Eval | 91.91 | 85.04 | 60.82 | 99.77 | 75.57 | 89.93 | 64.20 | 60.82 | 94.74 | 74.08 |
| | TPT | 91.89 | 85.55 | 61.13 | 99.78 | 75.81 | 89.93 | 64.41 | 61.16 | 94.83 | 74.36 |
| | TPT-C | 81.64 | 67.53 | 59.88 | **99.82** | 74.86 | 58.48 | 71.72 | 37.11 | 69.00 | 48.26 |
| | ROSITA | **99.10** | **7.63** | **72.81** | 99.74 | **84.17** | **94.79** | **32.59** | **66.64** | **96.40** | **78.80** |
| MAPLE | ZS-Eval | 98.48 | 3.77 | 72.08 | 99.60 | 83.63 | **98.34** | **7.86** | 73.08 | 97.58 | 83.57 |
| | TPT | 98.15 | 5.67 | 69.04 | 99.64 | 81.56 | 98.34 | 7.89 | 71.78 | 97.63 | 82.73 |
| | TPT-C | 98.56 | **3.74** | 71.87 | 99.64 | 83.51 | 98.32 | 8.18 | 72.76 | 97.87 | 83.47 |
| | PAlign | 98.15 | 5.67 | 70.02 | 99.64 | 82.24 | 98.34 | 7.90 | 72.95 | 97.64 | 83.51 |
| | PAlign-C | 98.56 | 3.74 | 71.84 | 99.65 | 83.49 | 98.32 | 8.13 | **78.71** | 97.89 | 83.46 |
| | ROSITA | **99.34** | 5.22 | **78.02** | **99.93** | **87.63** | 97.80 | 13.15 | 73.49 | **98.49** | **84.17** |

Table 23: Detailed results using CIFAR-10C as $D_d$ with Tiny ImageNet and CIFAR-100C as $D_u$.

| | Method | CIFAR-10C/Tiny | | | | | CIFAR-10C/CIFAR-100C | | | | |
|---|---|---|---|---|---|---|---|---|---|---|---|
| | | AUC | FPR | $Acc_D$ | $Acc_U$ | $Acc_{HM}$ | AUC | FPR | $Acc_D$ | $Acc_U$ | $Acc_{HM}$ |
| CLIP | ZS-Eval | 91.33 | 27.07 | 70.55 | 79.20 | 74.63 | 82.57 | 67.92 | 60.81 | 79.45 | 68.89 |
| | TPT | 91.31 | 27.23 | 71.55 | 79.17 | 75.17 | 82.57 | 68.06 | 61.15 | **79.61** | 69.17 |
| | TPT-C | 74.08 | 61.45 | 37.65 | 73.89 | 49.88 | 61.45 | 94.30 | 34.54 | 69.31 | 46.10 |
| | ROSITA | **96.43** | **12.10** | **74.81** | **86.11** | **80.06** | **82.99** | 62.89 | **66.63** | 72.75 | **69.56** |
| MAPLE | ZS-Eval | 90.86 | 27.54 | 74.49 | 77.66 | 76.04 | 86.14 | **52.08** | 67.99 | 75.97 | 71.76 |
| | TPT | 90.86 | 27.61 | 73.47 | 77.56 | 75.46 | 86.15 | 52.14 | 66.61 | **75.87** | 70.94 |
| | TPT-C | 91.18 | 26.93 | 75.27 | 77.37 | 76.31 | 86.50 | 50.56 | 70.59 | 71.56 | 71.07 |
| | PAlign | 90.86 | 27.60 | 74.49 | 77.53 | 75.98 | 86.15 | 52.18 | 67.65 | 75.85 | 71.52 |
| | PAlign-C | 91.18 | 26.90 | 75.28 | 77.35 | 76.30 | 86.50 | 50.58 | 70.58 | 71.51 | 71.04 |
| | ROSITA | **91.67** | **25.31** | **76.69** | **78.67** | **77.67** | **86.82** | 50.33 | **72.96** | 73.35 | **73.15** |

Table 24: Detailed results using ImageNet-C as $D_d$ with MNIST and SVHN as $D_u$.

| | Method | ImageNet-C/MNIST | | | | | ImageNet-C/SVHN | | | | |
|---|---|---|---|---|---|---|---|---|---|---|---|
| | | AUC | FPR | $Acc_D$ | $Acc_U$ | $Acc_{HM}$ | AUC | FPR | $Acc_D$ | $Acc_U$ | $Acc_{HM}$ |
| CLIP | ZS-Eval | 93.39 | 55.52 | 26.14 | 99.89 | 41.43 | 85.89 | 72.91 | 26.10 | 93.78 | 40.83 |
| | TPT | 93.12 | 58.01 | 26.76 | 99.88 | 42.21 | 85.43 | 74.47 | 26.18 | 94.03 | 40.95 |
| | TPT-C | 56.57 | 99.12 | 3.25 | 62.57 | 6.19 | 11.38 | 100.00 | 4.03 | 35.16 | 7.24 |
| | ROSITA | **99.52** | **4.06** | **32.04** | **99.97** | **48.53** | **98.34** | **10.21** | **30.21** | **99.21** | **46.32** |
| MAPLE | ZS-Eval | 81.49 | 92.95 | 26.60 | 96.40 | 41.70 | 83.26 | 71.15 | 28.06 | 89.81 | 42.77 |
| | TPT | 81.38 | 93.17 | 25.17 | 96.33 | 39.92 | 83.18 | 71.52 | 26.50 | 89.93 | 40.93 |
| | TPT-C | 83.25 | 87.60 | 27.55 | 95.96 | 42.81 | 83.18 | 70.60 | 28.28 | 88.49 | 42.86 |
| | PAlign | 81.38 | 93.17 | 26.30 | 96.33 | 41.32 | 83.18 | 71.52 | 27.65 | 89.93 | 42.30 |
| | PAlign-C | 71.22 | 86.32 | 16.78 | 70.89 | 27.14 | 32.17 | 94.32 | 10.36 | 30.29 | 15.44 |
| | ROSITA | **99.56** | **1.66** | **34.50** | **99.92** | **51.30** | **98.68** | **5.09** | **34.05** | **98.95** | **50.67** |

Table 25: Detailed results using CIFAR-100C as $D_d$ with MNIST and SVHN as $D_u$.

| | Method | CIFAR-100C/MNIST | | | | | CIFAR-100C/SVHN | | | | |
| | | AUC | FPR | $Acc_D$ | $Acc_U$ | $Acc_{HM}$ | AUC | FPR | $Acc_D$ | $Acc_U$ | $Acc_{HM}$ |
|---|---|---|---|---|---|---|---|---|---|---|---|
| CLIP | ZS-Eval | 77.78 | 99.93 | 32.05 | **98.68** | 48.39 | 64.70 | 98.68 | 32.05 | 80.55 | 45.85 |
| | TPT | 77.76 | 99.94 | 32.00 | 98.72 | 48.33 | 64.71 | 98.63 | 32.00 | 80.85 | 45.85 |
| | TPT-C | 51.57 | 100.00 | 17.51 | 59.31 | 27.04 | 9.40 | 99.98 | 3.62 | 13.90 | 5.74 |
| | ROSITA | **96.07** | **19.28** | **40.63** | 97.41 | **57.34** | **82.09** | **64.64** | **32.59** | **92.32** | **48.17** |
| MAPLE | ZS-Eval | 87.43 | 64.19 | 38.73 | 94.69 | 54.97 | 92.98 | 40.51 | 39.54 | 98.45 | 56.42 |
| | TPT | 87.42 | 64.09 | 36.89 | 94.68 | 53.09 | 92.97 | 40.44 | 37.55 | 98.48 | 54.37 |
| | TPT-C | 87.65 | 63.08 | 38.90 | 94.68 | 55.14 | 93.09 | 40.30 | 39.43 | 98.49 | 56.31 |
| | PAlign | 87.42 | 64.11 | 37.75 | 94.68 | 53.98 | 92.97 | 40.48 | 38.51 | 98.48 | 55.37 |
| | PAlign-C | 88.25 | 57.31 | 39.75 | 92.99 | 55.69 | 93.45 | 39.39 | 40.58 | 97.95 | 57.39 |
| | ROSITA | **97.04** | **11.01** | **45.11** | **99.41** | **62.06** | **96.26** | **20.99** | **42.30** | **98.89** | **59.25** |

Table 26: Detailed results using CIFAR-100C as $D_d$ with Tiny ImageNet and CIFAR-10C as $D_u$.

| | Method | CIFAR-100C/Tiny | | | | | CIFAR-100C/CIFAR-10C | | | | |
| | | AUC | FPR | $Acc_D$ | $Acc_U$ | $Acc_{HM}$ | AUC | FPR | $Acc_D$ | $Acc_U$ | $Acc_{HM}$ |
|---|---|---|---|---|---|---|---|---|---|---|---|
| CLIP | ZS-Eval | 67.31 | 73.89 | 35.35 | 65.01 | 45.80 | 63.28 | 93.25 | 32.04 | 70.42 | 44.04 |
| | TPT | 67.28 | 73.82 | 35.55 | 64.88 | 45.93 | 63.26 | 93.20 | 31.99 | 70.57 | 44.02 |
| | TPT-C | 59.74 | 79.76 | 10.68 | 66.75 | 18.41 | 55.86 | **86.35** | 7.64 | 63.33 | 13.64 |
| | ROSITA | **83.55** | **50.76** | **45.69** | **71.91** | **55.88** | **68.54** | 89.71 | **36.92** | 68.52 | **47.98** |
| MAPLE | ZS-Eval | 68.80 | 74.35 | 38.44 | 64.74 | 48.24 | 66.93 | 87.94 | 33.45 | 73.94 | 46.06 |
| | TPT | 68.80 | **74.20** | 36.88 | 64.65 | 46.97 | 66.93 | 87.95 | 31.75 | 73.71 | 44.38 |
| | TPT-C | 68.85 | 74.71 | 38.84 | 64.67 | 48.53 | 66.97 | 87.94 | 34.01 | 72.48 | 46.30 |
| | PAlign | 68.80 | 74.23 | **37.78** | 64.64 | 47.69 | 66.93 | 87.93 | 32.56 | 73.66 | 45.16 |
| | PAlign-C | 68.76 | 78.12 | 37.31 | 67.87 | 48.15 | 66.82 | 87.80 | 35.72 | **68.74** | 47.01 |
| | ROSITA | **70.37** | 77.00 | 37.62 | **68.97** | **48.68** | **69.57** | **83.61** | **38.03** | 68.09 | **48.80** |

Table 27: Detailed results using ImageNet-R as $D_d$ with MNIST and SVHN as $D_u$.

| | Method | ImageNet-R/MNIST | | | | | ImageNet-R/SVHN | | | | |
| | | AUC | FPR | $Acc_D$ | $Acc_U$ | $Acc_{HM}$ | AUC | FPR | $Acc_D$ | $Acc_U$ | $Acc_{HM}$ |
|---|---|---|---|---|---|---|---|---|---|---|---|
| CLIP | ZS-Eval | 91.27 | 91.09 | 55.67 | 99.90 | 71.50 | 90.43 | 75.04 | 56.36 | 98.38 | 71.66 |
| | TPT | 91.25 | 91.23 | 56.26 | 99.90 | 71.98 | 90.43 | 74.98 | 57.22 | 98.40 | 72.36 |
| | TPT-C | 82.81 | 85.79 | 51.86 | 99.78 | 68.25 | 80.94 | 80.03 | 54.88 | 93.55 | 69.18 |
| | ROSITA | **99.44** | **4.29** | **71.73** | **99.99** | **83.53** | **98.62** | **9.08** | **67.90** | **99.61** | **80.75** |
| MAPLE | ZS-Eval | 90.15 | 83.54 | 59.79 | 98.51 | 74.42 | 92.74 | 65.70 | 61.20 | 99.24 | 75.71 |
| | TPT | 90.14 | 83.58 | 59.26 | 98.51 | 74.00 | 92.74 | 65.68 | 60.56 | 99.26 | 75.23 |
| | TPT-C | 90.35 | 81.49 | 60.20 | 98.52 | 74.73 | 92.79 | 65.20 | 61.03 | 99.26 | 75.59 |
| | PAlign | 90.14 | 83.58 | 60.11 | 98.51 | 74.66 | 92.74 | 65.68 | 61.48 | 99.26 | 75.93 |
| | PAlign-C | 92.20 | 59.70 | 60.72 | 98.88 | 75.23 | 93.54 | 54.59 | 61.12 | 99.33 | 75.67 |
| | ROSITA | **99.39** | **2.95** | **73.49** | **99.96** | **84.70** | **97.85** | **12.98** | **71.14** | **99.80** | **83.07** |

