# OpenReview forum: "Efficient Open-world Test Time Adaptation of Vision Language Models"
_ICLR.cc/2025/Conference — ICLR 2025 Conference Withdrawn Submission_

### Official Review · Reviewer_fpUP · 2024-10-29

**Soundness:** 2
**Presentation:** 3
**Contribution:** 2
**Rating:** 5
**Confidence:** 5

**Summary:**

This paper proposes ROSITA, a new approach for Open World Single Image Test Time Adaptation that utilizes Vision-Language Models (VLMs). This method enhances the differentiation between known and unknown classes through the use of feature banks and a novel contrastive loss function.

**Strengths:**

**[New setup]** This work claims that it is the first to study open world single image test time adaptation using VLMs.

**[Extensive experiments]** This work has conducted extensive experiments across a diverse array of domain adaptation benchmarks.

**Weaknesses:**

**[Incremental technical contribution]** The main contribution is the reduce loss, a specific contrastive loss, while the class identifier seems a reformulation of previous methods [1, 2] to this task. Given this fact, the technical contribution seems incremental.

[1] Ronald A Fisher. The use of multiple measurements in taxonomic problems. Annals of eugenics, 7 (2):179–188, 1936.

[2] Yushu Li, Xun Xu, Yongyi Su, and Kui Jia. On the robustness of open-world test-time training:
Self-training with dynamic prototype expansion. In ICCV, 2023.

**[Unconvincing updating parameter choices]** In section 2.3, the authors test using three different parameter groups for training. However, only these three groups are not comprehensive. Other parameters might be more suitable for updating at test time, e.g., the first layer or the last layer of the encoders. Moreover, additional parameters to keep the intact of the original model are also worth trying, e.g., adapter layers and LoRA.

**[Unclear illustration]** In Figure 1, there might be some questions to be raised: 1) Which dataset is used for experiments? Is it representative enough? If not, better use the results on multiple datasets. 2) Is the learning rate only hyperparameter to tune? What about optimizer type, training epochs and learning rate schedular? 3) Is it possible to update both the prompts and the LN parameters?

**[Missed baseline]** The more recent paper “Diverse Data Augmentation with Diffusions for Effective  Test-time Prompt Tuning, ICCV 2023” is not compared.

**[Experiments]** (i) In Table 1 and 3, the proposed method often outperforms other methods significantly while in Table 2 the performance is close to that of (K+1)PC. Could the authors explain this? (ii) In Table 4, the experimental results with $L_{Re}$ and $L_D$ should be given. (iii) In Table 6, it would be nice to include other methods.

**Questions:**

What’s the main difference between CNN based open world TTA and VLMs based one?

---

> ### Author Response · Authors · 2024-11-20
> **Response to Reviewer fpUP (1/3)**
>
> We thank reviewer fpUP for the insightful comments and appreciating the new problem setup and extensive experimental analysis done in our work. We address your concerns as follows.
>
> ### **[W1]** **On incremental technical contribution**
> This being a common concern, we request reviewer m4nU to read the common response where we address this.
>
> ### **[W2+W3] Extensive preliminary analysis**
>
> Our initial experiments showed that updating LayerNorm parameters with simple entropy objective can effectively improve closed-set TTA performance. To illustrate this, we present our analysis for TTA of VLMs on CIFAR-10C dataset (lines 176-177). Further, based on the suggestions, we perform more extensive experiments **(i) Additional parameter choices:** (a) First attention block, (b) Last attention block of the Vision Transformer and (c) Both prompts and LayerNorm parameters. **(ii) Datasets:** In addition to CIFAR-10C, we experiment with ImageNet-R, a relatively large scale dataset consisting of 30,000 images from 200 classes. **(iii) Optimizer:** Along with SGD, we experiment with AdamW optimizer also used in [1], with varying learning rates on both CIFAR-10C and ImageNet-R dataset. We consistently observe that LayerNorm parameters is in general, a good choice to update the model.
>
>
> **Table 1:** Accuracy on CIFAR-10C with SGD optimizer and varying learning rates.
>
> | Parameters      | 1e-6  | 1e-5  | 1e-4  | 1e-3      | 1e-2    |
> |-------------|-------|-------|-------|-----------|---------|
> | Prompts     | 73.4  | 31.04 | 12.53 | 11.18     | 10.19   |
> | Full        | 10.48 | 10.44 | 9.99  | 10.0      | 10.01   |
> | First Block | 75.10 | 76.12 | 78.27 | 13.07     | 10.01   |
> | Last Block  | 73.45 | 72.42 | 59.44 | 10.17     | 10.02   |
> | Prompts+LN  | 73.82 | 46.77 | 24.71 | 10.24     | 10.18   |
> | LayerNorm   | 74.35 | 76.61 | 80.41 | **84.58** | 11.69   |
>
>
> **Table 2:** Accuracy on CIFAR-10C with AdamW optimizer and varying learning rates.
>
> | Parameters      | 1e-6  | 1e-5  | 1e-4     | 1e-3  | 1e-2  |
> |-------------|-------|-------|----------|-------|-------|
> | Prompts     | 72.4  | 18.6  | 12.83    | 10.04 | 10.08 |
> | Full        | 10.32 | 10.03 | 10       | 10    | 9.97  |
> | First Block | 79.05 | 24.7  | 10.84    | 10    | 10    |
> | Last Block  | 59.23 | 10.84 | 10.49    | 10    | 10.01 |
> | Prompts+LN  | 75.01 | 72.1  | 21.92    | 13.33 | 10.01 |
> | LayerNorm   | 76.1  | 81.57 | **85.9** | 85.27 | 10.03 |
>
>
> **Table 3:** Accuracy on ImageNet-R with SGD optimizer and varying learning rates.
>
> | Parameters      | 1e-6  | 1e-5  | 1e-4  | 1e-3      | 1e-2  |
> |-------------|-------|-------|-------|-----------|-------|
> | Prompts     | 73.97 | 74.17 | 74.71 | 25.68     | 10.63 |
> | Full        | 14.18 | 7.19  | 0.65  | 0.65      | 0.42  |
> | First Block | 73.84 | 74.31 | 74.91 | 8.76      | 0.32  |
> | Last Block  | 75.95 | 77.93 | 24.82 | 0.52      | 0.67  |
> | Prompts+LN  | 73.76 | 75.09 | 76.35 | 28.72     | 11.74 |
> | LayerNorm   | 74.13 | 74.35 | 75.23 | **76.92** | 33.07 |
>
>
> **Table 4:** Accuracy on ImageNet-R with AdamW optimizer and varying learning rates.
>
> | Parameters  | 1e-6  | 1e-5  | 1e-4  | 1e-3      | 1e-2  |
> |-------------|-------|-------|-------|-----------|-------|
> | Prompts     | 74.4  | 75.17 | 27.93 | 6.82      | 4.37  |
> | Full        | 14.83 | 0.95  | 0.28  | 0.52      | 0.66  |
> | First Block | 74.6  | 74.8  | 5.68  | 0.26      | 0.15  |
> | Last Block  | 77.44 | 10.67 | 0.51  | 0.25      | 0.33  |
> | Prompts+LN  | 74.52 | 76.45 | 12.99 | 8.87      | 5.55  |
> | LayerNorm   | 73.96 | 75.64 | 78.28 | **78.81** | 31.47 |
>
> As suggested, additional fine-tuning methods incorporating adapters, LoRA can definitely be explored although it increases the number of parameters. As this is not the primary focus of this work, we explore to choose the best subset of parameters in the existing model.

---

> > ### Author Response · Authors · 2024-11-20
> > **Response to Reviewer fpUP (2/3)**
> >
> > ### **[W4] Additional baseline**
> > Thanks for pointing us to this baseline[2]. However, we adapt a more recent, efficient and stronger baseline TDA (CVPR'24) [3] in our work.  Based on the complexity analysis presented in Table 2 in [3], DiffTPT takes _34h 45mins_ (excluding the image generation process with diffusional models) while TDA only takes about _16 mins_ for the same task. The testing time for DiffTPT is orders of magnitude huge compared to ZS-Eval and TDA, making it impractical to use in real world as low latency is a key consideration for TTA. Hence, we choose the more recent baseline TDA[3] in our work. In addition to TDA, we now adapt another very recent TTA method DPE (NeurIPS'24) [4] for Open-world TTA and report the results below. In DPE, they update visual and text prototypes through learnable residual parameters with an alignment and entropy loss. Although, DPE[4] outperforms TDA[3] in closed-set TTA, in the presence of undesired class samples, it's performance is worse than TDA[3]. This suggests that the prototype evolution with residuals as the learnable parameters can negatively hurt the model performance in open world scenario. We will revise the manuscript including DPE as another baseline.
> >
> > **Table 1:** Performance (HM) of ROSITA in comparison with DPE[4] on CIFAR-10C as $D_d$ with other $D_u$.
> >
> > | Method | MNIST     | SVHN      | Tiny      | CIFAR-100C |
> > |--------|-----------|-----------|-----------|------------|
> > | TDA    | 77.06     | 76.64     | 75.94     | 70.13      |
> > | DPE    | 27.60     | 68.54     | 69.90     | 62.34      |
> > | ROSITA | **84.17** | **78.80** | **80.06** | **69.56**  |
> >
> > **Table 2:** Performance (HM) of ROSITA in comparison with DPE[4] on CIFAR-100C as $D_d$ with other $D_u$.
> >
> > | Method   | MNIST     | SVHN      | Tiny      | CIFAR-10C |
> > |----------|-----------|-----------|-----------|-----------|
> > | TDA      | 46.52     | 46.01     | 47.52     | 45.79     |
> > | DPE      | 42.54     | 35.69     | 42.80     | 42.21     |
> > | ROSITA   | **57.34** | **48.17** | **55.88** | **47.98** |
> >
> > ### **[W5] Experiments:**
> >
> > (i) We modify (K+1) PC [5] to perform TTA using VLMs as described in Appendix B.2. As both (K+1)PC and ROSITA update the vision encoder, they outperform all other methods in general. Further, we see that the performance of ROSITA is close to that of (K+1)PC for ImageNet-R (Table 1), VisDA and DomainNet datasets (Table 2) compared to corruption benchmarks(ImageNet-C, CIFAR-10C, CIFAR-100C). We attribute this behaviour to the quality of pseudo labels in the dataset. While (K+1)PC updates the model using prototypical cross-entropy loss for each test sample, in ROSITA, the model is updated using the proposed ReDUCe loss only using reliable test samples. Compared to corruption datasets, we observe the pseudo labels are more reliable in ImageNet-R, VisDA and DomainNet datasets. This along with updating the vision encoder can achieve good results using (K+1)PC. However, for corruption datasets, more test samples are confusing (between desired and undesired classes). Using such samples to update the model can hinder the objective of separating the desired and undesired class samples. In ROSITA, we show that reliably choosing test samples for model adaptation (Appendix C.5) is consistently effective across datasets.
> >
> > (ii) We request the reviewer to clarify the question so that we can address it.
> >
> > (iii) **Memory buffer:** Prior prompt tuning methods like TPT[1] do not require any memory buffer. TDA[4] requires a memory buffer of size $(|C_d|\times(3+2))\times F$ to store 3 features per desired class in the positive cache and 2 features per class in the negative cache. ROSITA requires a memory buffer of size $(|C_d|\times 5 + 64) \times F$ to store 5 features per desired class and 64 features for the undesired classes. Here, F is the feature dimension. Both, TDA and ROSITA are memory efficient, requiring only about 10MB of additional memory even for a hard dataset like ImageNet-C with 1000 desired classes.
> >
> > [1] Shu et al., " Test-time prompt tuning for zero-shot generalization in vision-language models", NeurIPS 2022.
> >
> > [2] "Diverse Data Augmentation with Diffusions for Effective Test-time Prompt Tuning", ICCV 2023.
> >
> > [3] Karmanov et al., "Efficient test-time adaptation of vision-language models", CVPR 2024.
> >
> > [4] Zhang et al., "Dual Prototype Evolving for Test-Time Generalization of Vision-Language Models", NeurIPS 2024.
> >
> > [5] Li et al., " On the robustness of open-world test-time training:
> > Self-training with dynamic prototype expansion", ICCV 2023.

---

> > > ### Author Response · Authors · 2024-11-20
> > > **Response to Reviewer fpUP (3/3)**
> > >
> > > **[Q1]** **Difference between CNN and VLM based TTA:** Conventional TTA works primarily based on CNNs, involve a source model(trained specifically for each dataset) on clean data, say CIFAR-10 and is test-time adapted to CIFAR-10C, its corrupted test set. Contrary to this, VLMs like CLIP has been trained on large scale (image, text) pairs. This enables CLIP to be used in an off-the-shelf manner to classify any dataset given the $C$ classes of interest. **The text-based classifiers can be obtained for free using the text prompts like "A photo of a {classname}.", enabling effective zero shot $C$-way classification.** This makes VLM a natural candidate for TTA and hence, we choose to explore its effectiveness for Open-world TTA.

---

> > > > ### Comment · Reviewer_fpUP · 2024-11-23
> > > > **Comments on the Retuttal**
> > > >
> > > > Thanks for the author's response. The added experiments show the effectiveness of the method. However, the main concern is the limited technical novelty, which is not addressed well by the rebuttal. Hence, I will keep my score.

---

> > > > > ### Author Response · Authors · 2024-11-25
> > > > > **Response to Reviewer fpUP**
> > > > >
> > > > > Thank you for acknowledging the effectiveness of the method. We have done our best to address all the concerns raised. We have included these discussions in the revised manuscript and refer the reviewer to the respective sections to review at ease.
> > > > >
> > > > > 1. **Extensive analysis on the choice of parameters** (Appendix C.7).
> > > > > 2. **Additional Baselines:** As suggested, we adapt a very recent closed-set TTA approach for VLM, **DPE**[4] (NeurIPS 2024) in our problem setting. Further, as suggested by Reviewer m4nU, we adapt a CNN based open-set TTA method **UniEnt**[6] (CVPR'24) to our problem setting. ROSITA consistently outperforms these two recent benchmarks, further reinforcing the effectiveness of our method.
> > > > > 5. Loss ablation of $L_{Re}+L_D$ and $L_{Re}+L_U$, comparison of Memory buffer required by ROSITA and prior methods is included in the revised manuscript.
> > > > > 4. **Novelty:** We are the first to explore open-set TTA of VLMs to the best of knowledge. While closed-set TTA has been well explored with significant research interest, Open-set TTA is a more challenging and realistic problem, which is less explored. We extensively analyse the recent VLM based TTA approaches (currently handling only closed-set scenarios), recent open-set TTA approaches (currently explored in vision only backbones/CNNs). We identify the strengths and drawbacks of both sets of approaches and establish a strong set of baselines for Open-set single image TTA using VLMs.
> > > > > The design choices for ROSITA are well motivated based on this analysis:
> > > > >     - **Choice of learnable parameters** (Section 2.3 and Appendix C.7)
> > > > >     - **Dynamic thresholding based class identifier** (Appendix C.4)
> > > > >     - **ReDUCe loss**, specifically designed test-time objective addressing the challenges of Open-set Single Image TTA of VLMs (Table 2, Appendix C.3, C.4)
> > > > >     -  **Effectiveness of ROSITA in varied scenarios**:
> > > > >             (i) Test streams: Continuously, frequently changing domains, Varying ratio of $C_d/C_u$. (Table 5)
> > > > >             (ii) Light-weight TTA compared to prior prompt-tuning based methods. (Table 6 and Appendix C.8)
> > > > >             (iii) Performance agnostic to VL model architecture. (Tables 1,2,3 and Appendix C.9, D.2)
> > > > >             (iv) Consistent performance gains across domain shifts including corruption and a variety of style shifts. (Tables 1,2,3 and Appendix D.1)
> > > > >
> > > > > We strongly believe our work would be of interest to researchers exploring the adaptation capabilty of VLMs, as we take a step ahead of closed-set TTA and explore open-set TTA using VLMs. We hope this response addresses your concerns.
> > > > >
> > > > > [6] Z. Gao et al. "Unified Entropy Optimization for Open-Set Test-Time Adaptation", CVPR 2024.
> > > > >
> > > > > **As the discussion period is ending soon, we request you to go through the revised manuscript to verify the suggested changes and the additional experimental analysis included. We'd be happy to address any remaining concerns or provide further clarifications.**

---

### Official Review · Reviewer_m4nU · 2024-11-02

**Soundness:** 2
**Presentation:** 2
**Contribution:** 2
**Rating:** 3
**Confidence:** 4

**Summary:**

The article proposes ROSITA, a framework for test-time adaptation of vision language models in settings where, during inference, the model receives samples of both desired/target classes and undesired ones. The latter should be flagged by the model, as in standard open-set scenarios. ROSITA is based on two components: LDA-based separation of desired and undesired classes (following Li et al. (2023)), and a contrastive objective, updating LayerNorm parameters of the visual encoder. Experiments show that the approach outperforms common TTA approaches in this new, less-explored setting.

**Strengths:**

1. Although it is hard to define closed vs. open set in the context of VLMs, the case of test-time adaptation when the test samples comprise both desired and undesired classes is an understudied problem that this paper aims to tackle.

2. The proposed method, ROSITA, is sound, exploiting a simple technique based on LDA to identify desired vs undesired samples (Li et al. (2023)) coupled with a contrastive objective to refine the feature representation. The various design choices are also motivated via ablation studies (e.g., Fig. 1, Tab. 4).

3. The method consistently performs well across various settings and data streams.

**Weaknesses:**

1. As first statement for the focus of the work, lines 64-65 states this work adapts VLMs to work with single-image TTA. This sentence is misleading as multiple works already considered TTA with a single image: for instance, TPT (Shu et al. (2022)) and TDA (Karmanov et al. (2024)) are two examples. This sentence should be revised to avoid potential over claims and better contextualize the work, e.g., by focusing the claim on the differences in the specific setting considered and those of existing works on VLMs.

2. While the definition of the setting as open-world follows previous work (i.e., Li et al. (2023)), it is inaccurate as it does not follow its original meaning. Specifically, open-world recognition implies that classes are added overtime to the model, adding semantic to samples of unknown (or undesired, as in this context) classes [a]. This is not the case in this work, as the set of target classes is static and it is not updated. A more precise definition would be following (Lee et al. (2023)) and use "open-set TTA", as that denotes the presence of a set of target classes and OOD samples to be recognized. It would be helpful to clarify the particular meaning of open world (comparing it with previous use, e.g., [a]) and/or consider adopting the term "open-set TTA" throughout the paper for consistency with existing literature.

2. Currently, the set of baselines do not include any open-set TTA approach, beyond the adaptation of Li et al. (2023). In particular, there exist many potential techniques for detecting unknowns (e.g., MSP [b], max logic [c]) and previous works considered what happens when considering them as alternative OOD detection strategy (e.g., Lee et al. (2023), Tab. 5). As all the experiments and methods are held out with the same, LDA-based selection strategy (Appendix B.4, lines 809) it would be more thorough to include (i) alternatives to LDA's-based OOD detection (ii) simple combination of existing OOD identification algorithms and those for TTA. This would provide a more comprehensive evaluation of ROSITA's performance relative to a broader range of approaches.

4. Technically, the article heavily relies on the TDA scores (proposed already in Li et al. (2023)) to recognize undesired classes. At the same time, there has been already efforts in constructing contrastive-based objectives for TTA [d,e]. Tuning only LayerNorm layers is an already known concept in the literature [f] as well, reducing the impact of Sec. 2.3). In this context, the article merges these two approaches for improved TTA in the "open-world" setting but the technical contribution and the differences with what has been already presented should be clarified and potentially analysed in the manuscript. I would be helpful to include a paragraph to explicitly discuss how the approach differs from and improves upon existing contrastive TTA methods and LayerNorm tuning techniques, potentially highlighting the aspects (and challenges) that are especially relevant for open-set/world TTA.

6. TDA, one of the closest baseline, is compared with the proposed approach only on Table 3 and the same goes for the other main baseline, i.e., Li et al. (2023). Ii is not clear what is the criterion behind excluding part of the baselines between tables, especially two main ones which could further strengthen the contribution of the manuscript itself. TDA and Li et al. (2023) should be consistently reported across all relevant experiments, and/or it should be explained why these baselines have been excluded from certain comparisons. This would help readers better understand the relative performance of ROSITA across different scenarios.

**Minors**:
- From the abstract (i.e., lines 15-17) and the introduction (lines 73-75, 81-84) it is not clear how ROSITA work and what are its technical contribution. Clarifying them would allow the reader to better follow one of the key messages of the manuscript.

- Lines 203-205, the meaning of "text hypothesis transfer" is not clear.

- Line 481, typo "identifierr"

**Rerefences**:

[a] Bendale, Abhijit, and Terrance Boult. "Towards open world recognition." In CVPR, 2015.

[b] Hendrycks , Dan and Gimpel, Kevin. "A baseline for detecting misclassified and out-of-distribution examples in neural networks". In ICLR, 2017.

[c] Sungha Choi, et al. "Improving test-time adaptation via shift-agnostic weight regularization and nearest source prototypes." ECCV, 2022

[d] Chen, Dian, et al. "Contrastive test-time adaptation." In CVPR, 2022.

[e] Yamashita, Kota, and Kazuhiro Hotta. "MixStyle-Based Contrastive Test-Time Adaptation: Pathway to Domain Generalization." In CVPR, 2024.

[f] Zhao, Bingchen, et al. "Tuning LayerNorm in Attention: Towards Efficient Multi-Modal LLM Finetuning." In ICLR, 2024.

**Questions:**

1. Is the article the first to perform TTA on single samples and with OOD/undesired classes? If not, clarifying the differences would help to better understand the technical contribution.

2. How does the performance of the model change w.r.t. various design choices? (e.g., different OOD criterion, hyperparameters).

3. Is there a motivation behind the choice of competitors per method? And would it be possible to add other baselines as competitors?

---

> ### Author Response · Authors · 2024-11-20
> **Response to Reviewer m4nU (1/2)**
>
> We thank reviewer m4nU for the insightful comments. It is encouraging to hear that you appreciate our motivation for the problem, method and the experimental analysis presented. We address your concerns as follows.
>
> **[W1]** In line 64-65, our aim was to concisely describe the problem of our interest, Open world Single Image TTA. We clarify that we do not claim to be the first to do single image TTA. In fact, we acknowledge the current works TPT [1] and TDA [2] that do single image TTA (line 61) and adapt them as baselines for Open world single image TTA problem (line 153). We will revise this sentence based on your suggestion.
>
> **[W2]** We use the term "open-world" only to be consistent with the closely related work [3]. We clarify that, in this problem, we aim to correctly classify desired class samples and recognize others as undesired classes. Thank you very much for this suggestion, we will revise the manuscript adopting "open-set TTA" to be more consistent with the literature in general.
>
> **[W3]** We choose the LDA based class identifier, as we believe it rightly fits in the context of online adaptation as it estimates the threshold in an online manner. Determining any kind of threshold, for MSP, MaxLogit is not trivial in this setting as we do not have access to abundant data apriori. Hence, we adopt the LDA based detector [3]. As alternative choices, we experimented with MSP score, the results of which is presented in Appendix Section C.4 and Table 11 and here as well.
>
> Table : MSP vs LDA base class identifier with ReDUCe loss.
>
> | $\tau_u, \tau_t, \tau_d$ | CIFAR-10C | CIFAR-100C | ImageNet-C | ImageNet-R |
> |--------------------------|-----------|------------|------------|------------|
> | 0.4/0.6/0.8              | 43.44     | 34.42      | 1.20       | 77.12      |
> | 0.3/0.5/0.7              | 33.70     | 32.60      | 1.74       | 80.29      |
> | 0.5/0.5/0.5              | 22.82     | 37.41      | 1.91       | 30.90      |
> | ROSITA (LDA)             | 84.17     | 57.34      | 48.53      | 83.53      |
>
> The threshold $\tau_u$ is used to select reliable undesired class samples ($s_t<\tau_u$), $\tau_d$ is used to select reliable desired class samples ($s_t>\tau_u$) and $\tau_t$ is used to distinguish desired and undesired class samples. The experiments are done with each of CIFAR-10C,CIFAR-100C, ImageNet-C, ImageNet-R as desired class dataset $D_d$ with MNIST as $D_u$ (undesired class dataset).
>
> The first three rows in the above Table correspond to simple thresholding of MSP scores, where the thresholds are predefined and kept fixed throughout Test-time adaptation done using ReDUCe loss. We observe that the performance significantly varies for different choice of MSP thresholds suggesting that it is not desirable to choose these thresholds apriori in a TTA task as the MSP scores depend on several unknown factors like the type and severity of domain shift, confusion of classes etc.
>
> The score bank in LDA based identifier provides us with an estimate of the distribution of scores with time, based on which the threshold $\tau_t$ is continuously estimated. Apart from desired vs undesired class detection, the score statistics from LDA offer more information ($\mu_d, \mu_u$), which we leverage in ROSITA to identify reliable samples for TTA. From the above Table, we observe that LDA based class identifier used in ROSITA clearly outperforms fixed thresholding of MSP scores, justifying its effectiveness.
>
> **[W4] On technical contribution:**
> This being a common concern, we request reviewer m4nU to read the common response where we address this. Although we did attempt to highlight the differences in existing works and ours in the paper (Lines 477-502), we will refine this section to clarify these aspects further based on the suggestions.
>
> **[W5]** We clarify that TDA (Karmanov et al. (2024)) and (K+1) PC (adaptation of Li et al) is consistently compared across Tables 1-3 with CLIP backbone. Thank you for the suggestion, we will update the manuscript including results of TDA and (K+1)PC results for MAPLE VL backbone and also in Table 5 where we evaluate performance across different TTA scenarios as well. We are currently running these experiments.
>
>
> [1] Shu et al., " Test-time prompt tuning for zero-shot generalization in vision-language models", NeurIPS 2022.
>
> [2] Karmanov et al., "Efficient test-time adaptation of vision-language models", CVPR 2024.
>
> [3] Li et al., " On the robustness of open-world test-time training: Self-training with dynamic prototype expansion", ICCV 2023.
>
> [4] Lee et al., "Towards Open-Set Test-Time Adaptation Utilizing the Wisdom of Crowds in Entropy Minimization", ICCV 2023

---

> > ### Author Response · Authors · 2024-11-20
> > **Response to Reviewer m4nU (2/2)**
> >
> > We will revise the manuscript with the following minor changes as suggested.
> >
> > **[M1]** Thanks for the suggestion. We will refine these lines to provide better clarity.
> >
> > **[M2]** In lines 203-205, we intend to say, we transfer the text classifier &#40;hypothesis&#41; to test domains. The classifier decision boundaries defined by the text embeddings is kept fixed throughout the test time adaptation process. We will refine these lines to better convey the point.
> >
> > **[M3]** Thanks for pointing the typo. We will correct it.

---

> > > ### Comment · Reviewer_m4nU · 2024-11-22
> > > **Thank you for the response, some concerns remain**
> > >
> > > I thank the authors for responding to my concerns and those of the other reviewers. I still have concerns regarding the technical novelty and the competitors. Namely, the additional baselines still the baselines comparison is limited: i.e., while [4] should be added as per the reply to fpUP29, other open-set approaches and combinations of TTA ones for VLM with those (e.g., [g] below) are missing. Expanding the analyses and set of baselines is crucial to clearly state the need for specific open-set TTA solutions for VLMs. This is particularly needed as, when looking at the mode challenging settings (i.e., D_u as ImageNet or CIFAR-100C/CIFAR10-C, Tab. 3), the gap with the baselines is reduced: e.g., TDA outperforms the proposed approach on 2/3 metrics on CLIP with CIFAR-10C as D_u, the gap with MAPLE as backbone are all low (Tiny-ImageNet).
> > >
> > > Regarding the adaptation, the choice of LayerNorm tuning is agnostic to the actual open-set objective and the LDA-based separation is taken from [1]. Thus, the main technical contribution is the ReDUCe loss. As a contribution, it is of course valuable, but it would be interesting to check whether this objective alone boosts the performance of other TTA strategies in this setting.
> > >
> > > Finally, an updated version of the manuscript will confirm the implementation of the suggested changes.
> > >
> > > **References**
> > > [g] Z. Gao et al. "Unified Entropy Optimization for Open-Set Test-Time Adaptation", CVPR 2024.

---

> > > > ### Author Response · Authors · 2024-11-25
> > > > **Response to Reviewer m4nU (1/2)**
> > > >
> > > > Thanks for the thoughtful response. We address the remaining concerns here.
> > > >
> > > > **Baselines:** We include two additional baselines, UniEnt[g] (Thanks for pointing us to this work!) and DPE. UniEnt (CVPR'24) [g] is a very recent open-set TTA method for Vision only models/CNNs. DPE [5] is a very recent method proposed for closed-set single image TTA method of VLMs. We adopt both these methods for our problem, further strengthening our experimental analyses. **We include all these results in the revised manuscript.**
> > > >
> > > > [5] Zhang et al., "Dual Prototype Evolving for Test-Time Generalization of Vision-Language Models", NeurIPS 2024
> > > >
> > > > **Comparison with open-set TTA methods:**
> > > > To study the effectiveness of the proposed ReDUCe loss across different class identifiers, we equip ReDUCe loss with MSP based thresholding, Distribution Aware Filter (DAF) proposed in UniEnt[g] and the LDA based classifier. We present this analysis in detail in Appendix C.4 in the revised manuscript and also summarize it below.
> > > >
> > > > UniEnt employs a Distribution Aware Filter (DAF) by modeling similarity scores of each batch of test samples as a two-component Gaussian mixture Model (GMM). Here, as we only have a single sample at each time instant, we use a score bank (as done in LDA) to estimate the parameters of GMM to distinguish between desired and undesired class samples. UniEnt has three loss components, entropy minimization for desired class samples, entropy maximization for undesired class samples and maximization of average batch entropy to prevent mode collapse. As we have only one sample at a time, batch-wise entropy maximization is not applicable in our case. If the class identifier (DAF/LDA) recognizes a test sample $x_t$ to belong to $C_d$, then entropy minimization is employed, otherwise entropy maximization is employed. LayerNorm parameters are updating, as done is ROSITA. So, this experiment purely assesses the performance of the two loss functions in Open-set single image TTA scenario.
> > > > We equip the two Class Identifiers, DAF and LDA with two test time adaptation objectives, UniEnt[g] and ReDUCe loss and report the results below.
> > > >
> > > > Table: LDA/DAF class identifiers with UniEnt and ReDUCe loss on four $D_d$ with MNIST as $D_u$.
> > > > | Method     | CIFAR-10C | CIFAR-100C | ImageNet-C | ImageNet-R |
> > > > |------------|-----------|------------|------------|------------|
> > > > | LDA+UniEnt | 75.62     | 48.31      | 41.53      | 71.73      |
> > > > | DAF+UniEnt | 79.43     | 50.12      | 46.52      | 79.30      |
> > > > | LDA+ReDUCe | **84.17** | **57.34**  | **48.53**  | **83.53**  |
> > > > | DAF+ReDUCe | 83.56     | 55.37      | 48.33      | 83.32      |

---

> > > > > ### Author Response · Authors · 2024-11-25
> > > > > **Response to Reviewer m4nU (1/2)**
> > > > >
> > > > > We observe that ReDUCe loss significantly outperforms the Unified Entropy objective proposed in [g]. While UniEnt performs well in the case of CNNs and given access to a batch of samples, it has limitations when employing it to single image TTA using VLMs:
> > > > >
> > > > > 1.  Both LDA and DAF were proposed for CNN based open-set TTA where a source model is trained on say clean data and is adapted to new domains, with the observation that the feature-prototype similarity scores $s_t$ can distinguish desired and undesired class samples. In the case of VLMs, the source model is trained on a large scale dataset and is adapted to potentially unseen/corrupted/covariate-shifted data. _The prior that the feature-prototype similarity scores $s_t$ can distinguish desired and undesired class samples does not translate to VLMs as the scores overlap significantly_, as observed in ZSEval histogram plots in Figures 3,5,6 in the main paper.
> > > > > 2. In the case of CNNs, where the the initial scores are well separated and model has access to a batch of test samples at a time, UniEnt leverages this to further aid the separation of desired and undesired class samples in the batch through the UniEnt objective. In the case of VLMs, the scores are not well separated initially. This results in the means $\mu_d$ and $\mu_u$ in the case of LDA to be very close leading to misclassification of $C_d$ and $C_u$ class samples using the estimated threshold $\tau_t$.  Similarly, in the case of DAF, the two components of GMM would not be very distinctive to well distinguish desired and undesired class samples. This misclassification can result in entropy minimization being applied on $C_u$ samples and entropy maximization on $C_d$ samples, which is undesirable. Employing UniEnt objective with several misclassified samples may not actually separate desired and undesired classes, as also empirically observed in Tables 1,2,3 (UniEnt has high FPR rate in general). Entropy maximization of $C_u$ samples does not explicitly enforce the separation of desired and undesired class samples in the feature space.
> > > > > 3. The $L_D$ and $L_U$ loss components of ReDUCe loss explicitly enforce the separation desired and undesired class samples in the common VL latent space, while the $L_{Re}$ loss aims to only align the desired class samples to align with the text prototypes. With time, the model is adapted such that undesired class samples are away from the desired class samples and also the text prototypes. This ReDUCe loss addresses the challenges in single image open-set TTA in a holistic manner, resulting in better performance.
> > > > > 4. On adopting UniEnt objective to single-image TTA, either entropy minimization or maximization loss would be active based on whether a test sample is identified as desired or undesired class sample, which is a limitation, as the objective cannot enforce distinction between the two types of features.
> > > > > 5. In the case of CNNs, where the the initial scores are well separated and model has access to a batch of test samples at a time, UniEnt leverages this to further aid the separation of desired and undesired class samples in the batch through the UniEnt objective. In the case of VLMs, the scores are not well separated initially, hence the ReDUCe loss components (with the help of feature banks) is the driving force to better separate the desired and undesired class samples in the common latent space, resulting in lower FPR rates as a consequence.
> > > > >
> > > > >
> > > > > These observations suggest that the assumptions of the problem, (1) CNNs/Vision only backbones vs VLMs, (2) Batch vs single sample TTA, can have significant impact on the performance of a method. It is not trivial to translate the effectiveness of a method in one problem setting (batch-wise, CNNs) even to a closely related yet different problem setting (single-sample, VLMs). This motivates the need for designing test-time objectives specifically for VLMs to adapt using single samples in open-set scenarios.
> > > > >
> > > > > Our work was very much driven by this motivation to analyse open-set TTA methods, closed-set VLM based methods to identify their drawbacks, and then design a well motivated solution for open-set TTA of VLMs. Thanks to the constructive feedback during the review process, we have included two recent strong baselines to compare against our method, ROSITA.
> > > > >
> > > > > **We request you to go through the revised manuscript to verify the suggested changes and the additional experimental analysis included. As the discussion period is ending soon, we'd be happy to address any remaining concerns or provide further clarifications.**

---

> > > > > > ### Author Response · Authors · 2024-11-28
> > > > > > **Remaining concerns addressed**
> > > > > >
> > > > > > Dear reviewer,
> > > > > >
> > > > > > We have done our best to address the remaining concerns raised by you, in the previous responses and the revised manuscript. The main concern being, lack of sufficient baselines, based on the suggestions, we have added two strong baselines in the revised version of our manuscript: 1. We adopt **DPE**(NeurIPS’24) which is a closed-set single image TTA method (along the lines of TDA (CVPR’24)) to open-set single image TTA. 2. Thanks for referring us to **UniEnt** (CVPR’24) which is an open-set TTA method proposed for batch-wise TTA of  vision only backbones/CNNs. We employ UniEnt to perform single image open-set TTA of VLMs.
> > > > > >
> > > > > > We believe the latest revision of our work has a comprehensive set of baselines to compare our method against and justifying its effectiveness. Please let us know if you have any further questions, so that we can answer them while we still can. We would welcome any constructive feedback, that can help improve this work, leading toward its acceptance.
> > > > > >
> > > > > > Looking forward to hearing from you.

---

### Official Review · Reviewer_Pk3P · 2024-11-02

**Soundness:** 3
**Presentation:** 3
**Contribution:** 3
**Rating:** 6
**Confidence:** 5

**Summary:**

This paper propose a novel method called ROSITA  for Open World Single Image Test Time Adaptation using VLMs.

ROSITA employs feature banks and an innovative contrastive loss to enhance the distinction between known and unknown classes, allowing efficient adaptation to domain shifts while enabling the model to reject unfamiliar classes.

The extensive experiments demonstrate the effectiveness of the proposed method.

**Strengths:**

1. The paper is well-written and easy to follow, with a clear motivation.
2. The experimental results are impressive, and the ablation study is thorough.
3. The method is novel, interesting, and effective in addressing the challenging problem of Open World Test Time Adaptation.

**Weaknesses:**

1. The title of this paper uses '...... vision-language models' , It seems that author only used CLIP for experiments. It would be better to also study the performance with CLIP-large and other VL backbones.
2. The paper lacks a thorough discussion of the limitations of the current solution and does not provide suggestions for future work.
3. The author doesn't report the error bar.

**Questions:**

Please see the weakness.

---

> ### Author Response · Authors · 2024-11-20
> **Response to Reviewer Pk3P**
>
> We sincerely thank reviewer Pk3P for the valuable feedback provided. It is encouraging to know that you appreciate several aspects of our work. We now address your concerns as follows.
>
> ### **[W1] Experiments with more VL backbones**
>
> All the main experiments were performed on two VL backbones, CLIP ViT-B/16 and MAPLE, which is an adaptation of CLIP incorporating visual and text prompts. To study the performance of our method across architectures,  experiments using CLIP-ViT-B/32 and CLIP-ResNet-50 VL backbones were done, results of which were reported in Appendix D.2 (Table 15-17). Further, as suggested, we perform experiments on CLIP-Large model ViT-L/14 and report the results here. **We observe that ROSITA outperforms prior methods even on large models like CLIP ViT-L/14 and OpenCLIP[1] ViT-L/14** (trained on LAION 2B dataset). We will revise the manuscript including these results.
>
> **Table 1(a):** Results with CIFAR-10C as $D_d$ (Desired) and others as $D_u$ (Undesired) using **CLIP-ViT-L/14** model.
>
> | Method  | MNIST | SVHN    | Tiny  | CIFAR-100C |
> |---------|-------|---------|-------|------------|
> | ZSEval  | 83.94 | 74.54   | 80.16 | 72.32      |
> | (K+1)PC | 85.43 | 80.60   | 81.65 | 71.90      |
> | TDA     | 84.91 | 76.87   | 81.07 | 74.23      |
> | ROSITA  | **89.46** | **83.42**   | **83.61** | **75.63**      |
>
>
> **Table 1(b):** Results with CIFAR-100C as $D_d$ (Desired) and others as $D_u$ (Undesired) using **CLIP-ViT-L/14** model
>
> | Method  | MNIST | SVHN    | Tiny  | CIFAR-10C |
> |---------|-------|---------|-------|-----------|
> | ZSEval  | 56.29 | 52.35   | 53.25 | 49.89     |
> | (K+1)PC | 64.14 | 55.18   | 54.53 | 47.90     |
> | TDA     | 59.11 | 55.25   | 55.44 | 52.48     |
> | ROSITA  | **65.41** | **60.31**   | **57.55** | **54.66**     |
>
>
> **Table 2(a):** Results with CIFAR-10C as $D_d$ (Desired) and others as $D_u$ (Undesired) using **OpenCLIP-ViT-L/14** model.
>
> | Method  | MNIST | SVHN  | Tiny  | CIFAR-100C |
> |---------|-------|-------|-------|------------|
> | ZSEval  | 80.64 | 76.90 | 84.10 | 75.40 |
> | (K+1)PC | 85.84 | 82.42 | 84.99 | 75.70 |
> | TDA     | 80.57 | 77.92 | 84.60 | 75.79 |
> | ROSITA  | **89.04** | **82.98** | **85.55** | **76.62** |
>
>
> **Table 2(b):** Results with CIFAR-100C as $D_d$ (Desired) and others as $D_u$ (Undesired) using **OpenCLIP-ViT-L/14** model
>
> | Method  | MNIST | SVHN    | Tiny  | CIFAR-10C |
> |---------|-------|---------|-------|-----------|
> | ZSEval  | 62.96 | 59.38   | 61.10 | 59.57     |
> | (K+1)PC | 70.14 | 63.36   | 60.56 | 59.43     |
> | TDA     | 64.90 | 60.70   | 62.01 | 61.20     |
> | ROSITA  | **70.54** | **63.84**   | **62.57** | **61.84**     |
>
> [1] M. Cherti et al. "Reproducible scaling laws for contrastive language-image learning", CVPR 2023.
>
>
> ### **[W2]**  **Limitations and Scope for Future work**
>
> Although ROSITA performs better than the baselines, in datasets where CIFAR-10C is $D_d$ and CIFAR-100C is $D_u$, where it is hard to distinguish desired and undesired class samples, the FPR is still quite high, indicating that there is still significant scope for improvement. While in this work, we aim to identify the undesired class samples as "I don't know", in many practical applications these new classes can be of interest and need to be included in the desired classes. The desired class set keeps growing in this, which is more challenging. This incremental nature of TTA can be potentially explored in the future.
>
> The Limitations were discussed in the Appendix. As suggested, we will include the above discussion in the manuscript revision.
>
>
> ### **[W3] Error bars**
> The performance of ROSITA is analysed for five random seeds, generating differently ordered test data, the results reported in Appendix Section C.1 (Table 8).  As we report several metrics for each dataset, we omit this in the main Tables for better readability.

---

> > ### Comment · Reviewer_Pk3P · 2024-11-26
> >
> > I would like to thank the authors for their thoughtful responses to my concerns. Their responses have adequately addressed my main issues, and as a result, I will maintain my original score

---

> > > ### Author Response · Authors · 2024-11-28
> > > **Thank you for the positive response**
> > >
> > > Dear reviewer,
> > >
> > > We are glad to hear that our answers have addressed your concerns. It would be great if you could soon let us know if there are any further clarifications required, while we can still answer them. We would really appreciate it if you could specify anything in particular that was not satisfactory or any suggestions in general, to further improve our work, leading to its acceptance.
> > >
> > > Thanks again for your thoughtful response. Looking forward to hearing from you.

---

### Author Response · Authors · 2024-11-20
**Authors Response on common concern (1/2)**

**Regarding Incremental Novelty:**
The driving motivation for this work was to establish a strong benchmark for Open world TTA. Here, we attempt to make a convincing case about why we believe this work is a step forward in advancing research in TTA. This work progressed by us questioning and answering the following sequence of questions, in this order.

1. **Why is Open-set/world TTA a relevant problem?** TTA is a very relevant and important problem as it equips the model to adapt to the ever-changing domains in the real world. Most TTA works study the closed-set scenario where the test samples come from a predefined set of classes. Well, real world is seldom closed. On encountering an unfamilar object (open classes), a system equipped to perform closed-set classification would misclassify this unfamiliar object as one of the predefined closed set of classes. A system that identifies a bicycle (say unfamiliar object) as car (in closed-set) is not very smart. A system with the ability to say "I don't know" is better than the one that misclassifies a bicycle as car. Hence, we believe it is important to explore beyond the conventional closed set TTA to bridge the gap between research in TTA and real world TTA. Open-set/world TTA has been recently explored using TTA based on CNNs [1,2].
2. **Why use VLMs?** Conventional TTA works primarily based on CNNs [7, 10], involve a source model (trained specifically for each dataset), say trained on CIFAR-10 clean data and is test time adapted to CIFAR-10C, its corrupted test set. Contrary to this, VLMs like CLIP has been trained on large scale (image, text) pairs. This enables CLIP to be used in an off-the-shelf manner to classify any dataset given the $C$ classes of interest without any specific retraining. _The text-based classifiers can be obtained for free using the text prompts like "A photo of a {classname}.", enabling effective zero shot $C$-way classification._ This makes VLM a natural candidate for TTA [3,4,5] and hence, we choose to explore its effectiveness for Open-world TTA.
3. **How do we setup the baselines?**
Recent works like TPT [3], PromptAlign [4], TDA [5] show that VLMs can be test-time adapted to improve results beyond zero-shot classification on a single-image basis. These however only address closed-set TTA. To adapt a method from closed-set to open-set, we need an additional module that distinguishes desired class samples from the undesired class samples. For this purpose, we adapt the LDA based OOD detector proposed in [1] for single image open-world TTA here. In our work, _we combine the single image TTA works on VLMs [3,4,5] with LDA based class identifier [1] to establish a strong benchmark for Open world Single Image TTA._
4. **Why use LDA based class identifier?**  This can be done by thresholding scores like MaxLogit, MSP[6]. However, it is non-trivial to choose a fixed threshold, especially in this single image TTA scenario where we do not have access to any/abundant data. In [1], they present a very simple and elegant LDA based OOD detector to estimate the threshold dynamically during TTA, which we adapt for our work. _We emphasize that we do not claim this to be our contribution, and we use it across all the baselines presented._ In ROSITA, we leverage the score statistics which we get for free from the LDA based class identifier to select reliable samples for updating the model using ReDUCe loss.
5. **What is the best choice of parameters to update the model?** There is a clear gap between CNN [7,10] and VLM [3,4,5] based TTA works with respect to the parameter choice, which we attempt to bridge by presenting this analysis (Section 2.3).  As updating normalization layers has been successfully used in many CNN based TTA works[7], it is only natural to consider adapting normalization layers even in the case of VLMs. We believe Entropy Minimization [7] with _LayerNorm update_ [8] is the most natural and simplest baseline to initiate the study of TTA of VLMs. Surprisingly, none of the current TTA works [3,4,5,9] on VLMs establish this simple yet strong baseline (which is obvious in hindsight). The motivation to present this analysis is to establish this baseline, emphasizing that learnings from TTA works based on CNNs can indeed be translated to VLMs. Yes, we agree this is not novel. Nevertheless, we believe it is an important insight, and hope future VLM based TTA works can benefit from this.

---

> ### Author Response · Authors · 2024-11-20
> **Authors Response on common concern (1/2)**
>
> 6. **What can be a good loss objective for Open world single image TTA?**
> The goal here is to separate undesired class samples from desired class samples while performing well on desired classes. A major contribution of our work is to design this ReDUCe loss, specifically to address the challenges in open-world TTA. Contrastive loss [12, 13] is a very powerful tool, successfully used in several applications  across few-shot to abundant data scenarios (too many to cite). While all the contrastive objectives [10, 11, 12, 13] look similar to Eqn.(6-7), they still differ based on how it is adapted in the context of a given problem. For e.g, AdaContrast[10] adapts Momentum Contrastive (MoCo) [12] loss to TTA, while MoCo [12] itself is adapted from SimCLR [13]. So, yes, mathematically the ReDUCe objective takes a form similar to several prior works [10,11,12,13]. Nevertheless, we believe its design for this problem is a major technical contribution of our work. We motivate our design choices for open-world single image TTA as follows. (i) ReDUCe loss components: The losses $L_D$ and $L_U$ both aim to _separate desired and undesired class samples_. As we aim to classify the desired class samples into one of the $|C_d|$ desired classes, we design $L_D$ to specifically drive features from the same desired class samples to cluster together and the clusters to be far apart. (ii) We design two _dynamically updating feature banks_ from which the positives and negatives for $L_D$ and $L_U$ are picked from. (iii) The feature banks are populated only with _reliable samples_, the need for such careful selection is ablated in Appendix C.5. (iv) The role of each loss component is clearly justified through the Ablation presented in Table 4.
>
>
> [1] Li et al. " On the robustness of open-world test-time training:
> Self-training with dynamic prototype expansion", ICCV 2023.
>
> [2] Lee et al. "Towards Open-Set Test-Time Adaptation Utilizing the Wisdom of Crowds in Entropy Minimization", ICCV 2023
>
> [3] Shu et al. " Test-time prompt tuning for zero-shot generalization in vision-language models", NeurIPS 2022.
>
> [4] Hassan et asl. "Align Your Prompts: Test-Time Prompting with Distribution Alignment for Zero-Shot Generalization", NeurIPS 2023..
>
> [5] Karmanov et al. "Efficient test-time adaptation of vision-language models", CVPR 2024.
>
> [6] Hendrycks et al. "A baseline for detecting misclassified and out-of-distribution examples in neural networks". In ICLR, 2017.
>
> [7] Wang et al. "Tent: Fully Test-time Adaptation by Entropy Minimization", ICLR 2021.
>
> [8] Zhao, Bingchen, et al. "Tuning LayerNorm in Attention: Towards Efficient Multi-Modal LLM Finetuning.", ICLR 2024.
>
> [9] Zhang et al., "Dual Prototype Evolving for Test-Time Generalization of Vision-Language Models", NeurIPS 2024.
>
> [10] Chen, Dian, et al. "Contrastive test-time adaptation", CVPR 2022.
>
> [11] Yamashita, Kota, and Kazuhiro Hotta. "MixStyle-Based Contrastive Test-Time Adaptation: Pathway to Domain Generalization", CVPR 2024.
>
> [12] He, Kaiming et al. "Momentum Contrast for Unsupervised Visual Representation Learning", CVPR 2020.
>
> [13] Chen et al. "A Simple Framework for Contrastive Learning of Visual Representations", ICML 2020.

---

### Author Response · Authors · 2024-11-20
**Summary of the rebuttal**

Dear reviewers,

We sincerely thank you all for providing constructive feedback on our work. This has definitely helped strengthen our work from several perspectives.

Here, we summarize the major concerns raised by the reviewers' which we address during this rebuttal:

1. **Experiments with more Vision Language Backbones.**
2. **Justifying the choice of LDA based class identifier.**
3. **Regarding technical contributions and novelty of this work.**
4. **Extensive analysis on parameter choices for model update.**
5. **Additional baseline for TTA using VLMs.**

We are currently revising the manuscript based on the suggestions provided and including additional experimental results. In the meanwhile, **we look forward to actively discuss and address any further concerns through the discussion period.**

---

### Author Response · Authors · 2024-11-25
**Manuscript Revision**

**Dear Reviewers,**

Thank you for the insightful feedback, which has helped us significantly to strengthen our work. We have revised the manuscript to incorporate your suggestions and additional experimental analyses done as part of this review process.

Below, we summarize the major updates done in the manuscript:
1. **Comprehensive analysis of learnable parameters** (Section 2.3 and Appendix C.7).
2. Inclusion of **additional baselines**: **DPE** (NeurIPS'24) and **UniEnt** (CVPR'24) (Tables 1, 2, 3, and Appendix B).
3. **Evaluation of different $C_d$ vs $C_u$ class identifiers** (Appendix C.4).
4. **Key distinctions between ROSITA and prior works** (lines 391–410).
5. **Detailed study of the effectiveness of the ReDUCe loss** (lines 109–117).
6. **Performance evaluation of ROSITA with larger VL backbones** (Appendix C.9).
7. **Discussion of limitations and scope for future work** (Appendix B.5).

As the discussion phase is drawing to a close, we look forward to addressing any remaining questions or concerns you might have. Thank you once again for your time and valuable input!

---

> ### Author Response · Authors · 2024-11-27
> **Manuscript updates**
>
> Dear reviewers,
>
> We sincerely request you to go through the recently revised manuscript which includes the suggestions, experiments and discussion done through the review process. As today is the final day to submit a revised manuscript, I would like to inquire if there are any concerns regarding our manuscript that we could address through revisions to enhance its evaluation.

---

### Note · Authors · 2025-01-15

I have read and agree with the venue's withdrawal policy on behalf of myself and my co-authors.